# BEYOND LABELING ORACLES:
# WHAT DOES IT MEAN TO STEAL ML MODELS?

## ABSTRACT

Model extraction attacks are designed to steal trained models with only query access, as is often provided through APIs that ML-as-a-Service providers offer. ML models are expensive to train, in part because data is hard to obtain, and a primary incentive for model extraction is to acquire a model while incurring less cost than training from scratch. Literature on model extraction commonly claims or presumes that the attacker is able to save on both data acquisition and labeling costs. We show that the attacker often does not. This is because current attacks implicitly rely on the adversary being able to sample from the victim model's data distribution. We thoroughly evaluate factors influencing the success of model extraction. We discover that prior knowledge of the attacker, *i.e.* access to in-distribution data, dominates other factors like the attack policy the adversary follows to choose which queries to make to the victim model API. Thus, an adversary looking to develop an equally capable model with a fixed budget has little practical incentive to perform model extraction, since for the attack to work they need to collect in-distribution data, saving only on the cost of labeling. With low labeling costs in the current market, the usefulness of such attacks is questionable. Ultimately, we demonstrate that the effect of prior knowledge needs to be explicitly decoupled from the attack policy. To this end, we propose a benchmark to evaluate attack policy directly.

## 1 INTRODUCTION

Modern ML models are valuable intellectual property, in part because they are expensive to train (Sharir et al., 2020; Patterson et al., 2021), and model extraction (ME) attacks (also known as model stealing or black-box estimation) threaten their confidentiality. In a ME attack, the adversary attempts to steal a victim model over query access, in order to obtain an approximate copy of that model with similar performance (Tramèr et al., 2016). In the most common class of ME attacks, the adversary uses queries to the victim model as a training set for the copy of that model. Figure 5 illustrates such a typical attack, while we discuss other types of ME attacks in Section 2.1.

ME attacks are a looming threat for ML-as-a-Service providers, whose business model depends on paid query access to their models over APIs. In fact, Kumar et al. (2020) outline that ME is one of the most concerning threats for the industry. Typically, the claimed motivation to conduct ME attacks is to avoid the costs (see Appendix A) involved in training a model from scratch; the implicit assumption is that ME is data- and/or compute- cheaper (Jagielski et al., 2020). Here, the adversary's goal is to get the best performing model using the least resources. Current literature uses the ratio between the number of samples $M$ used to train the victim model and the number of queries $N$ used to train the stolen copy as a measurement of the attack's efficacy; If $M$ is much larger than $N$, the attack is a success. This criterion stems from the costs associated with data labeling and training.

Intuitively, this query efficiency criterion can be possible, provided the attacker can identify the most informative queries. However, we find that much of current research does not adequately account for how the adversary's prior knowledge contributes to the attacks' query efficiency. In many experiments, prior work assumes that the attacker has access to unlabeled data from a similar distribution to the victim's training data, but does not rigorously study how having this prior to the victim's training distribution contributes to the attack's efficiency (Correia-Silva et al., 2018; Orekondy et al., 2019a; Dziedzic et al., 2022a; Jagielski et al., 2020; Pal et al., 2020; Zhang et al., 2021; Okada et al., 2020; Karmakar & Basu, 2023). We refer to these as data access assumptions. In

contrast, when such priors are removed, the query complexity of *data-free model extraction* attacks drastically increases (Truong et al., 2021). This is important because, given enough prior knowledge of in-distribution data, it becomes cheaper for the adversary to train their own model from scratch without querying the victim. If all the adversary needs is labels for the unlabeled data they have access to, in many use cases it might be more cost-effective to obtain those using a crowdsourcing service, *e.g.* Amazon's Mechanical Turk, than by querying the victim model – as we discuss in Appendix A. This, in turn, removes their incentive to perform ME.

In this paper, we study how data access assumptions influence the accuracy of extracted models, and how this changes adversaries' incentives. First, we show that with the amount of prior knowledge (*i.e.* unlabeled datapoints) that many current works assume, for commonly used benchmarks one can train a model with comparable accuracy to that of the victim model. Second, we demonstrate that soft-labels only lead to minor improvements in accuracy of the stolen model ($\sim 4\%$) for cases with little prior knowledge, and the effect disappears with more data available. **In other words, aside from providing a label, victim labels leak limited information about the decision boundaries.**

Next, to better understand the adversary's dependence on prior knowledge of the distribution, we investigate a surprising phenomenon where models can sometimes get extracted with data that is unrelated to the model task (Orekondy et al., 2019a). We investigate the extent to which such out-of-distribution (OOD) queries expose the model's behaviour for in-distribution (IND) data. We suggest a novel benchmark with explicit control over the informativeness of OOD queries by instructing the model to give fake, yet plausible, decisions. We show that this causes both a drop of the stolen model accuracy and increases data complexity. Furthermore, by reducing informativeness of OOD data, we implicitly force an intelligent adversary to solve a problem of predicting if a given datapoint is IND or OOD, a problem known to be as hard as classification itself (Tramer, 2021). Put simply, to extract a model in this setting, an attacker needs to either have IND data, or a very large query budget. Yet, when such data is available, the attacker once more gets nothing but a label from the victim.

Overall, our paper finds that the practical threat posed by ME attacks is often exaggerated, since the attacker bears comparable costs to the victim. To summarize, we make the following contributions:

- We examine how prior knowledge of the victim model's training distribution contributes to the efficiency of ME. We show that if the adversary has access to even just a small amount of IND data, ME simply becomes a labeling oracle. In other words, aside from providing a label, victim labels leak limited information about the decision boundaries.

- We then investigate cases where OOD data is used for ME and find that IND data dominates extraction, yet OOD also leaks information about IND decision boundaries (Section 5).

- We modify victim models so that, for OOD queries, they output labels that are uncorrelated with IND decision boundaries, disrupting ME (Section 5). Our new benchmark controls for IND leakage from OOD queries and evaluates the attacker's knowledge of the distribution.

## 2  RELATED WORK

### 2.1  MODEL EXTRACTION ATTACKS

Model extraction attacks, also called model stealing, were first explored by Tramèr et al., who proposed an attacker that queries the victim model to label its dataset and trains a model to match these predictions. Various works extended the attack with the goal of achieving similar *task accuracy* while minimizing the number of queries required. Most of these attacks either assume access to a surrogate dataset (Correia-Silva et al., 2018; Orekondy et al., 2019a) or to a portion of the real training set (Rakin et al., 2021), or use random queries (Krishna et al., 2019; Chandrasekaran et al., 2020) in domains like NLP. We discuss the relation between ME to active learning in Appendix B. Truong et al. attested that, in order to successfully extract the victim model, the attacker's dataset must share semantic or distributional similarity to the real training dataset; otherwise resulting in an insufficient extraction accuracy. To mitigate this issue, they propose a data-free model extraction attack (DFME) that does not require a surrogate dataset. Inspired by data-free knowledge distillation (Lopes et al., 2017), they train a generative model to synthesize queries which maximize disagreement between the attacker's and the victim's models. A similar method was also proposed by MAZE (Kariyappa

et al., 2021). These results align with our analysis and show that the adversary can compensate for the absence of prior knowledge by using a very large query budget.

Jagielski et al. proposed a ME attack that, in addition to *accuracy*, targets *fidelity*. In this scenario, the attacker focuses on the exact reproduction of the victim model behaviour on all possible inputs. Carlini et al. argued that ME attacks are essentially a cryptanalytic problem, and proposed an attack based on differential cryptanalysis, with the goal of high fidelity extraction. Note that this line of research falls outside the scope of our work, since both the worst and the average-case query complexities reported by both papers significantly exceed accuracy-based ME attacks empirically. As such, in this work we will focus our evaluations on ME attacks that target task accuracy.

While the most commonly used threat model for ME attacks assumes black-box query access to the victim model, it is important to note that there is a significant line of work which focuses on side-channel ME attacks (Zhu et al., 2021; Hu et al., 2019; Yan et al., 2020; Hua et al., 2018; Xiang et al., 2020; O'Brien Weiss et al., 2023; Duddu et al., 2018). This line of work is out of scope for our paper since most side-channels are fixable with careful system re-design, while ME attacks should remain unaffected as long as model provides genuine responses to queries.

## 2.2 Defenses Against Model Extraction Attacks

Having established the vulnerability of ML models to ME attacks, we turn to defenses. It is worth mentioning that in the limit there is little that a victim can do to stop ME. Unlimited data sampling allows for exhaustive search of all possible inputs, whereas functions that are used to approximate decision boundaries in the limit can approximate arbitrary functions (Hornik et al., 1989). This leads to an inherent performance trade-off, where the victim sacrifices performance of the model to limit information leakage to an acceptable level. Broadly, defenses against ME attacks can be categorized as *active*, *passive*, *reactive*, or *proactive*, based on when they are applied in the extraction process.

**Active defenses** are applied to the model's predictions with the purpose of manipulating the prediction of the attacker's queries. This includes prediction poisoning (Orekondy et al., 2019b), where small perturbations are added to the model's predictions to poison the training objective of the attacker, and prediction truncation (Tramèr et al., 2016) These defenses come at the cost of lowering the quality of predictions for legitimate users. **Passive defenses** attempt to detect and block an attack (Juuti et al., 2019). **Reactive defenses** address extraction attacks post-hoc by determining if a suspect model was stolen or not. This includes watermarking (Jia et al., 2021a), dataset inference (Maini et al., 2021), and proof-of-learning (Jia et al., 2021b). Dziedzic et al. proposed a **proactive defense**, which attempts to deter the attack before it happens, by requiring the completion of a proof-of-work.

We argue that, although our OOD component degrades ME performance, it should not be considered a defense mechanism. Rather, we position it as a benchmark for validating our hypothesis regarding the adversary's assumption over the informativeness of the OOD region. It does, however, share a common intuition with many passive and active defenses, in the sense that they too target the amount of information that can be obtained from querying the victim model, across the entire input space.

## 3 Warm-up: Linear classification

In this section, we elaborate on our intuition behind the success of model extraction attacks, which later motivates the design of our benchmark to evaluate the impact of prior knowledge on ME attacks.

Consider a linear classification model that splits the input space into two decision regions, and the corresponding decision boundary is defined by the line $y = \alpha x$. Here, $(x, y)$ denotes the two-dimensional input, and $\alpha$ is the single model parameter that determines the entire decision boundary, which we demonstrate in Figure 1. ME in this context is equivalent to estimating the decision boundary over the green IND region where $x \sim [20, 80]$. The attacker is assumed to have the ability to query the model with arbitrary data but has no knowledge of where the IND region starts or ends. We assume that the depicted red region, $x \sim [0, 20] \cup [80, 100]$, is OOD. Note that it is only important to perform well on the task of interest in the IND region, and neither the victim model's owner nor the attacker are concerned with making predictions for OOD inputs. An attacker with no precise knowledge of the range of the input domain, *i.e.* what is IND and what is out of it, will require more parameters to approximate the whole range, since they do not know where the task begins and ends.

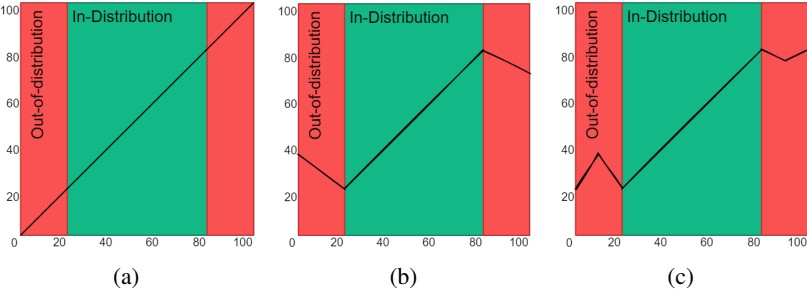

Figure 1: Consider a linear classifier for which the decision boundary is given by the line $y = \alpha x$. An attacker attempts to steal the model (*i.e.* find the corresponding $\alpha = 1$ from the example above). The green region $x \sim [20, 80]$ is in-distribution behaviour that the attacker wants to replicate, the red region $x \sim [0, 20] \cup [80, 100]$ is out of distribution and is not important for the task. The left case requires a single parameter to be approximated, the middle needs 5, whereas the right requires 9.

**The left-most plot** represents the case of a naive victim model with the original decision boundary. Here ME reduces to estimating a single parameter $\alpha$. Note that, in this case, the IND model can be extracted equally well using both IND or OOD data points, as the model behaviour is shared between both domains. **In the middle case**, we slightly increase the complexity of the decision boundary in the OOD area by introducing a new linear boundary with non-zero intercept in each region. As the attacker does not know where the IND and OOD regions are, it must approximate a piece-wise linear boundary, *i.e.* the slopes and the intercepts of the new lines, in addition to the slope of the original IND boundary. This increases the required number of parameters to 5 and, consequently, the cost of ME. **In the right-most case**, we further increase the complexity as before by introducing two additional linear decision boundaries per region, resulting in a total of 9 parameters to be estimated. The above example demonstrates that while the IND area was not modified, adding complex and task-independent boundaries to OOD regions can significantly increase the attacker's required capacity, thus making model extraction more costly. The same intuition also holds for the minimal required sample size. In the left-most case, the adversary needs at least 2 data points to estimate the parameter $\alpha$; in the middle case, the adversary needs at least 6 data points; and for the right-most case, at least 10 data points are required. The increase in the complexity of the decision boundary to be estimated results in an increase in both capacity and the minimal required sample size.

This behaviour is related to the locally independent nature of large models when performing regression or classification tasks. As sampling the boundary at one point does not necessarily reveal any information on the boundary IND, the attacker must explore the entire space. The main assumption here is that *the attacker can not know which areas are important*. In other words, we find that model extraction attacks implicitly assume that the adversary has prior knowledge of the distribution to be able to reproduce the victim model's predictions on the task of interest. Without such knowledge, the adversary is unable to tell whether making a query to the victim model will aid its goal of learning the true decision boundaries. In Section 5.2, we evaluate this assumption by equipping the victim model with an OOD detection component. In this setting, we find that the adversaries end up "wasting" queries on learning non-important boundaries that do not contribute to their goal.

Hence, we demonstrate that ME adversaries either need access to prior knowledge about the distribution or the ability to submit a large number of queries to the victim model. In the next sections we will extensively analyze this for real models and attacks, and show that this claim holds in real-world settings.

## 4 MODEL EXTRACTION ATTACKS AND PRIOR KNOWLEDGE

To evaluate the risk ME attacks pose, we measure the effect of the adversary's prior knowledge over the data distribution. Section 4.2 presents a baseline attacker that only utilizes its prior knowledge of the data distribution to train a model. Section 4.3 evaluates commonly used ME attacks and demonstrates that they only marginally improve upon the baseline attack accuracy.

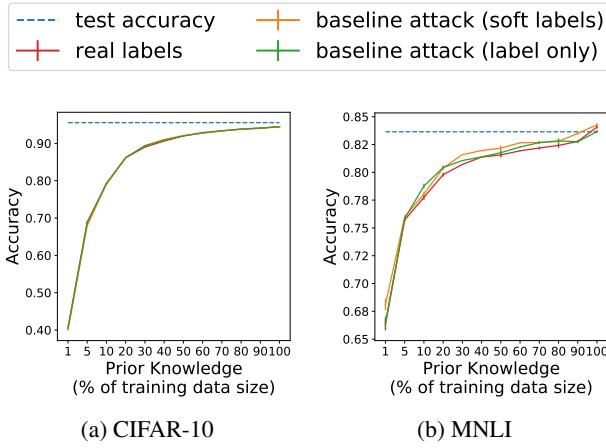

Figure 2: Evaluation of the risk posed by an attacker with some prior knowledge over the true data distribution. The prior knowledge is expressed as access to a percentage of the true training set. Attackers with more than 50% access can fully extract the model. However, in this case, the extraction is not more efficient than training from scratch. Attackers do not gain much by querying the victim model instead of using real labels, which also appears equivalent to label-only access to the victim. This shows that the victim is essentially a labeling oracle.

(a) CIFAR-10        (b) MNLI

## 4.1 EXPERIMENTAL SETTING

We evaluate ME attacks on both vision and NLP tasks. We measure the attacker's performance as the accuracy difference between the victim and the attacker on the original task. We define a successful attack as one that (i) produces a model that performs on a task nearly as well as the victim model and (ii) achieves this with a lower or equivalent sample complexity to that of training the victim model. Unless stated otherwise, in all of our experiments we assume that the attacker has a black-box access to the victim model: the adversary issues a query, and the victim model responds with a vector that indicates the probability of classifying the input in each of the task classes. We note that this setting provides the most information to the adversary, and we will explore more limited settings in which the adversary is limited to label-only access. Benchmarks are compatible with the ME literature.

**Vision.** For vision tasks, we evaluate the CIFAR-10 dataset (Krizhevsky et al., 2009), for which we follow the setting and training details described by the state-of-the-art DFME attack (Truong et al., 2021). We additionally evaluate the Indoor67 (Quattoni & Torralba, 2009), CUBS200 (Wah et al., 2011) and Caltech256 (Griffin et al., 2007) datasets, and follow the setting and training details described by the Knockoff Nets attack (Orekondy et al., 2019a), one of the strongest ME attacks. Due to space limitation, we provide the Knockoff Nets results in Appendix G, and focus the discussion on CIFAR-10. In all experiments we use the pretrained victim models provided by the authors.

**NLP.** We evaluate the MNLI classification task (Williams et al., 2018), in a standard setting (Krishna et al., 2019). For the victim model, we use the publicly-released, pre-trained 12-layer transformer BERT-base model (Devlin et al., 2018), and fine-tune it for 3 epochs with a learning rate of 0.00003.

## 4.2 BENCHMARKING MODEL EXTRACTION AND TRAINING

First, we evaluate our baseline attacker. Here, we assume the attacker has access to a randomly sampled subset of the victim's true training dataset, and uses this data only to query the victim and train the attack model. Figure 2 shows the attack accuracy as a function of the percentage of the data available to the attacker. Since ME quality is mainly affected by the quality of the attacker queries, real victim training data represents one of the best query sets for the attacker (compressed datasets *e.g.* (Wang et al., 2018) or disjoint IND data can potentially lead to an equally capable extraction set). Figure 2 shows that with only 10% of the data, the attacker already gets relatively high performance, *i.e.* 79.21% out of 95.54% for CIFAR-10 and 77.85% out of 83.64% for MNLI, and with 50% it approaches the original test accuracy of the model, *i.e.* 92.09% for CIFAR-10 and 81.94% for MNLI. An attacker with access to more than 50% of the data can extract the victim model successfully.

While the risk such an attacker poses is quite high, in practice, if the attacker has access to most of the victim training data, it could have already trained the model on its own, and at most the model serves as a labeling oracle. To better demonstrate this, in Figure 2 we compare the attack accuracy, using soft-label access to the victim model, to the accuracy the attacker would have obtained using real (ground truth) labels. Evidently, the attacker does not benefit much from attacking the model.

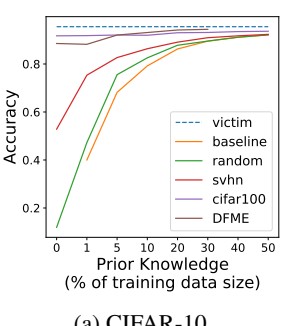
(a) CIFAR-10

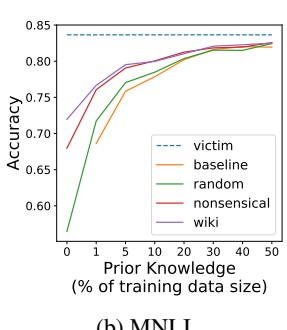
(b) MNLI

Figure 3: Effect of augmenting the attacker's queries with additional queries sampled from other data distributions, compared to our baseline attacker, which only uses its prior knowledge. We fix the query budget to be the size of the original training set for a fair comparison. Attackers with more prior knowledge over the true distribution do not benefit much by augmenting the query set. Moreover, as previously shown by Truong et al., the similarity between the victim's data distribution and the query distribution has a large impact on the attack's success.

We further demonstrate this point by comparing the attack accuracy when the attacker has label-only access. We observe that this case is equivalent to the real-label performance, indicating that, besides serving as a labeling oracle for the attacker, the extraction does not show much gain over simply training from scratch. This phenomena aligns with the prior literature (Orekondy et al., 2019b).

### 4.3 CAN SURROGATE DATASET AID EXTRACTION?

We now investigate the ME performance gains from the use of additional queries sampled from data distributions that differ from the victim model's training distribution, yet are 'similar' to it, following Truong et al.; Krishna et al.. Here, we consider three adversaries: the first assumes no prior knowledge; the second has partial knowledge about the (related) data; the third has unrestricted access to label any given data point. Namely, for CIFAR-10, we additionally use: **Random** - uniformly sampled random images; **Surrogate** - real images sampled from a surrogate (SVHN and CIFAR-100) dataset; **DFME** - synthetic queries generated using the DFME attack. For Indoor67, CUBS200, and Caltech256, we evaluate only the surrogate variant using the ImageNet dataset, as described by Orekondy et al.. Results are presented in Appendix G. For MNLI we use: **Random** - nonsensical sequence of random words built from the WikiText-103 corpus (Merity et al., 2017), sampled letter-by-letter; **Nonsensical** - nonsensical sequence of real words built from the WikiText-103 corpus, sampled word-by-word; **Wiki** - real sentences sampled from the WikiText-103 corpus.

We measure the improvement in attack accuracy as a function of the additional query budget and prior knowledge. We focus on the attacker with less than $50\%$ IND data access, since in Section 4.2 we show that higher prior knowledge renders attacks unnecessary. Due to the high computational cost of DFME, we only evaluate this type of queries for the attacker with up to $30\%$ prior knowledge.

We first investigate the case where the query budget is constant and is set to the size of the original training dataset, *i.e.* the maximal query complexity still considered a success. For each prior knowledge percentage $x\%$, we fill the remaining queries with any of the previously described methods. We include the case where all the queries are sampled from the query distribution, and the attacker has no ($0\%$) prior knowledge, to better visualize the effect of the prior knowledge. Note that DFME requires a query budget significantly higher than the size of the training set, so here we set it to 20M.

The results, presented in Figure 3, show that an attacker with more prior knowledge gains little benefit from additional queries. In the case of CIFAR-10, an attacker with $50\%$ prior knowledge only increases its success rate by up to $1.4\%$. For MNLI, this number is $0.6\%$. We do observe an improvement for the low-query setting, yet it is not enough to successfully extract the victim model – with the exception of the CIFAR-100 and the DFME cases. CIFAR-10 and CIFAR-100 are very similar datasets, with some classes being semantically similar *e.g.* automobiles and trucks. This is not the case for DFME, where the queries are synthesized, however, in this case, a very large query budget (20M) is used. The high budget can be explained by the intuition provided in Section 3, where in the absence of knowledge of the relevant regions to sample queries from, the adversary is required to traverse the input space and use a large query budget for this. In Appendix C we evaluate the effect of additional queries for a given prior knowledge percentage, showing diminishing returns from them.

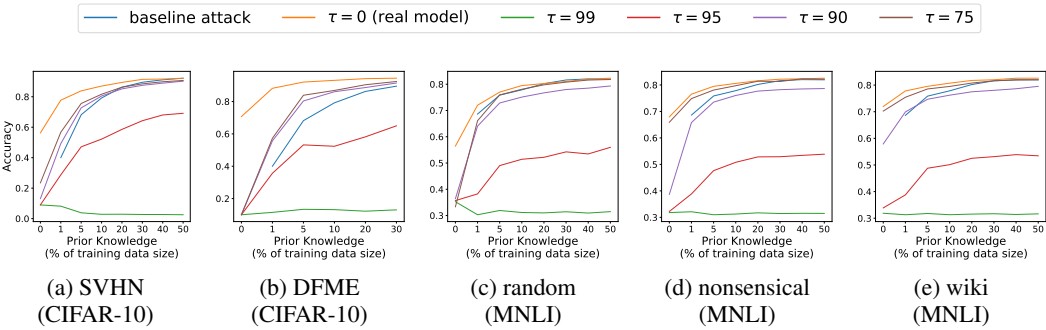

(a) SVHN  (b) DFME  (c) random  (d) nonsensical  (e) wiki
(CIFAR-10)  (CIFAR-10)  (MNLI)  (MNLI)  (MNLI)

Figure 4: The effect of applying our OOD component with different values of $\tau$ against an attacker that utilizes additional queries. In all cases other than DFME, the attacker adds $\|D_{train}\|$ additional queries, and for DFME adds 20M queries. When comparing the results to the original setting (real model), where the OOD region is unmodified, we can see a clear decrease in the attack accuracy.

We thus conclude that ME attacks need a formal definition of their adversarial goals – we observe that the adversary can achieve its goal of training an accurate model with just IND datapoints labeled, without extra queries to the victim model. We find that in all our experiments with different attacks, the victim model acts as a labeling oracle, and the adversary simply trains with labeled data.

## 5    LIMITING OUT-OF-DISTRIBUTION EFFECTIVENESS

Following our motivating example in Section 3, we explore the extent to which the attacker relies on an implicit assumption that the IND decision boundaries can be inferred from the OOD ones. If this was not the case, then an intelligent attacker would have to distinguish between IND and OOD samples, or otherwise waste queries and potentially learn non-existent decision boundaries. We augment our classifiers with OOD detection components. Similarly to the Figure 1(b-c), we modify the decision boundaries in the OOD region such that the adversary is unable to statistically tell IND or OOD responses apart. In this setting, we observe that attacks are no longer query effective (unless they have prior knowledge of the data distribution) because not all queries provide an informative response—and it is not easy for the adversary to tell which queries do.

### 5.1    SAMPLING COMPLEXITY INTUITION

When useful responses are limited to the IND region only, the attacker's success is dominated by the percentage of queries sampled from IND, which is a function of the attacker's prior knowledge of the victim's true distribution. In this scenario, the weakest attacker, with no prior knowledge, is the random guess attacker that samples useful information with probability $|\mathcal{X}_{useful}|/|\mathcal{X}|$, where $\mathcal{X}$ is the entire input domain and $\mathcal{X}_{useful}$ is the IND part of the domain. The strongest attacker here has precise knowledge of the IND region within $\mathcal{X}$ and can sample only from this region, which results in samples useful for the attacker's goal. This is the strongest attacker, as no ME defense can truly defend against such an attacker unless it corrupts the utility of the model as it predicts on the IND domain. Instead, the average-case attacker has partial prior knowledge about the true distribution. As such, the probability of sampling IND is a function of this prior, which indicates the overlap between the true distribution and the attackers' query distribution. We estimate the probability of randomly sampling IND for models considered in this section and find it equals $0.00001\%$ for CIFAR10 models and $0.01\%$ for MNLI. We expand on the procedure in Appendix E.

### 5.2    ON HARDNESS OF OOD DETECTION

By modeling the attacker's prior knowledge over the data distribution as a function of the distance between the victim's data distribution and the attacker's query distribution, we can see that the probability of sampling an informative IND datapoint decreases when this distance gets larger. As such, the attacker is inevitably sampling from both IND and OOD regions. Following the previously described assumption where the task-related information is limited to IND queries, the attack success

is now a function of the ability to solve a non-trivial OOD detection task: an adversary who is unable to distinguish between the IND and OOD samples inevitably faces a higher query complexity for a successful attack. Despite significant progress in OOD detection, the literature still remains largely empirical (Fang et al.; Hendrycks & Gimpel, 2016b; Carlini & Wagner, 2017) and finds that OOD detection is not at all trivial. Interestingly, robust OOD detection is linked to research into model robustness. Recently, Tramer demonstrated that robust adversarial example detection is as hard as classification. Here, robustness of the detector refers to consistent detection over the $\epsilon$-radius around a data point, representing the maximal distance for an adversarial example. Fundamentally, Tramer demonstrates that robust detection over an $\epsilon$ region around a given (not necessarily adversarial) data point implies, albeit inefficient, an ability to successfully classify for at least $\epsilon/2$ region.

Simply put into Bayesian interpretation, Tramer says that an ability to compute $\mathbb{P}_{\mathcal{D}}(x = x_i, y = y_i)$, *i.e.* perform OOD detection robustly with knowledge of the labels, implies an ability to classify robustly $\mathbb{P}_{\mathcal{D}}(y = y_i | x = x_i)$. Intuitively, that follows from the Bayes rule with $\mathbb{P}_{\mathcal{D}}(x = x_i, y = y_i) = \mathbb{P}_{\mathcal{D}}(y = y_i | x = x_i) * \mathbb{P}_{\mathcal{D}}(x = x_i)$, where $\mathbb{P}_{\mathcal{D}}(x = x_i)$ represents prior knowledge over the true data distribution $\mathcal{D}$, *i.e.* the likelihood of $x_i$ coming from $\mathcal{D}$. In this paper we do not make assumptions about the labels – one can certainly imagine an attacker who has access to some labeled data. Do note however that label information is not necessary for a classic out-of-distribution detection *i.e.* checking if a given point comes from a given distribution $\mathbb{P}_{\mathcal{D}}(x = x_i)$. There are two possibilities. First, if the attacker knows the labels of the points being queried, then the setting is exactly the same as the one considered by Tramer *i.e.* robust OOD detection is reducible to robust classification. Second, if the attacker is assumed to be capable of estimating $\mathbb{P}_{\mathcal{D}}(x = x_i)$ robustly, reduction to classification without labels is impossible – the attacker would simply need to use the model as a labeling oracle.

Practically, this implies that with the OOD component in place, ME might not be the most efficient way for the attacker to achieve their objectives, given the capabilities they have access to. If they have the labels and an OOD detector, then they can already perform classification. If they do not have the labels, they can at most get the benefit of labeling their dataset, after which they get classification. If they are unable to solve the OOD detection problem, they would have to first build an OOD detector (which means collecting costly datapoints), since otherwise OOD queries provide them no information about the IND region. In other words, once we take into account the OOD component in place, a **model extraction attacker is only as good as their knowledge of the underlying dataset; but if this knowledge includes labels, often there is no need to extract models in the first place.**

## 5.3 OUT-OF-DISTRIBUTION INSTRUMENTATION

In Sections 5.1 and 5.2, we discussed the intuition behind our hypothesis that most previous attacks make an implicit assumption about the usefulness of OOD queries. We showed that when this assumption does not hold, and the OOD region is not indicative of the IND behaviour, ME attack success reduces to mostly being a function of the level of prior knowledge, or an attacker's ability to solve a non-trivial OOD detection task. We also argued that at a point when an adversary has the ability to solve OOD task, the attacker has enough knowledge to train the model themselves, indicating that the victim model merely serves as a labeling oracle.

In this section we simulate a setting where this implicit assumption does not hold, and empirically show how the attack success is, again, only dependent on prior knowledge of the data distribution. For this evaluation, we revisit the attacker that utilizes both prior knowledge and additional OOD queries, described in Section 4.3. We now propose a simple modification to the victim model that targets the assumption over the OOD region and limits the informativeness of the OOD behaviour.

### 5.3.1 OOD IMPLEMENTATION DETAILS

We propose a simple method for invalidating the informativeness of the OOD data region. Given the original victim model $\mathcal{V}_o$, we create a hybrid victim model $\mathcal{V}_h$ by combining $\mathcal{V}_o$ with an additional module $\mathcal{V}_f$ with different, or additional, decision boundaries. This additional model will be used to provide predictions for OOD queries that differ from the predictions they would have gotten from the original model. For each query $x$, the hybrid model $\mathcal{V}_h$ applies some decision rule $R$ to classify $x$ as IND or OOD, and uses the corresponding model for prediction.

We design the additional model $\mathcal{V}_f$ such that the decision boundaries of both models would have similar smoothness properties; thus, learning the "fake" boundaries is expected to be of the same

level of difficulty, with nearly statistically indistinguishable output distributions. We discuss this further in Appendix F.4.

We now describe the implementation of the decision rule $R$ and the complex model $\mathcal{V}_f$, and expand on details in Appendix F. For the decision rule $R$, we apply some pre-defined threshold $\tau$ over the prediction confidence of $\mathcal{V}_o$, with a softmax temperature of 2 to better calibrate $R$. If the confidence is higher than $\tau$, the hybrid model returns $\mathcal{V}_o(x)$, otherwise it returns $\mathcal{V}_f(x)$. In Appendix F.3, we discuss the effect of $\tau$ in more detail. Prediction confidence serves as a naive OOD detector (Hendrycks & Gimpel, 2016a; DeVries & Taylor, 2018) and represents one of the simplest settings for the attacker. More advanced OOD detectors will strictly improve separation IND and OOD queries, therefore decreasing the number of false negative queries, *i.e.* OOD queries that are classified IND and are provided with a meaningful prediction by $\mathcal{V}_o$.

We provide the full description and details of the fake model $\mathcal{V}_f$ in Appendix F, and only give here a brief overview. We implement $\mathcal{V}_f$ by fitting a Gaussian Mixture Model (GMM) to the logits of each class, using the real training data samples. We additionally create anchor points $\mathcal{A}_c^1, \ldots, \mathcal{A}_c^m$ for each class, using feature representations of each class' data samples. These will be used to "assign" queries to one of the GMMs for prediction. The anchor points are permuted between the classes, to ensure that tail-of-distribution queries will not be "correctly" assigned to their real class, and therefore leak IND information. We denote this permutation by $\pi_j : C \to C$, for $j \in [m]$.

For a given query sample $x$, we compute its feature representation and find the nearest $L_2$ anchor point $\mathcal{A}_i^j$. We sample fake logits $\tilde{y}$ from the GMM for the chosen anchor point, according to its permutation $i' = \pi_j[i]$. We return it as the fake model prediction $\mathcal{V}_f(x) = \tilde{y}$. This mechanism results with OOD samples being "wrongly" predicted while still exhibiting similar smoothness properties.

### 5.3.2 OOD AND THE ADDITIONAL QUERIES

We show how our OOD component reduces the informativeness of OOD queries and thus the accuracy the attacker gained by utilizing additional OOD queries. We evaluate different threshold values ($\tau$) against an attacker that, in addition to its prior knowledge samples, uses additional $\|D_{train}\|$ queries, *i.e.* the size of the original training dataset. For DFME, we allow that attacker to use 20M additional queries. Different threshold values influence the results by changing the number of additional queries that are predicted by $\mathcal{V}_f$ versus $\mathcal{V}_o$ – the lower the threshold, the better for the attacker. We measure the false-positive rate (FPR) for the different threshold values in Appendix F.3.

We present the results of adding our OOD component in Figure 4. The results clearly demonstrate the described effect. As the attacker has less prior knowledge, it is more reliant on the benefit of the additional queries, and, as such, it is more affected by adding our OOD complex model $\mathcal{V}_f$. The attack accuracy is thus reduced closer to, or bellow, the accuracy of the baseline (which makes no additional queries). The effect of the OOD component is weaker for some of the settings in our NLP task, specifically the nonsensical and wiki queries. This is due to the similarity between the true data distribution and these query distribution. We discuss this observation in detail in Appendix H.

## 6    CONCLUSION

In this paper, we show that ME attacks can often be more expensive for the adversary than developing a model from scratch. We demonstrate that the performance of an attacker with a reasonable query budget is bounded by their access to IND data. Yet, given sufficient IND data, the victim model mainly serves as a labeling oracle, which in turn is not necessarily more cost-effective than online labeling services. Augmenting the IND data with data from another distribution relies on informative responses for task-irrelevant queries. We show that decorrelation of OOD from IND responses changes the attacker-victim dynamic, where attacks become much less cost-effective, forcing the attacker to collect IND data. Although ME attacks represent a serious threat in practice, our work highlights that corresponding risks can be reduced by making extraction expensive with an appropriate incentive mechanism design. We leave this implementation for future work.

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

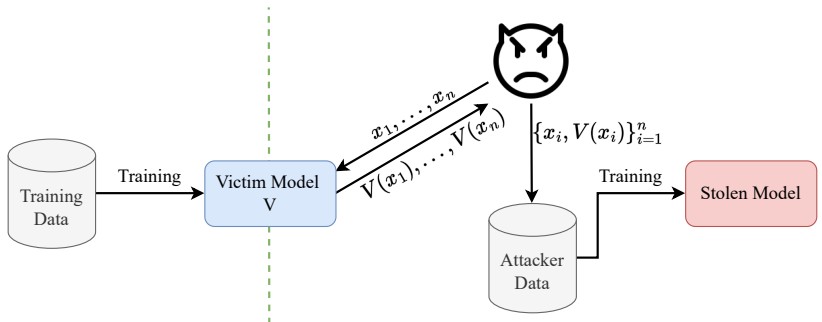

Figure 5: Illustration of model extraction attacks. In this setting, the victim model is trained on some training data that is not accessible by the adversary. The adversary attempts to steal the victim model over query access, in order to obtain an approximate copy of that model with similar performance. Typically, the adversary achieves this by querying the victim model to collect an "attacker dataset". This, in turn, is used to train a stolen, surrogate copy that mimics the victim's behavior.

## A PRIMER ON ML COSTING

Current model extraction literature never formally defines what makes a model extraction attack successful. While for high-fidelity extraction attacks an attacker explicitly expects to extract a model within a given error, accuracy-driven model extraction can be considered successful if stealing a model costs less than developing it.

The overall cost of producing and deploying a model can be broken down into six main parts:

**1** Data and Model storage      **4** Data labeling

**2** Training and Deployment hardware      **5** Model training

**3** Data collection      **6** Data and Model management

Out of the six, current model extraction really only focuses on data collection, labeling and training; training costs are usually only mentioned in passing and the vast majority of papers scale their attacks with just data collection. Although rationality of management and hardware costs in model extraction are indeed interesting, they fall outside of the current threat models in the literature.

Now we turn to CIFAR10 benchmark as a costing case study. Here we assume CIFAR10 dataset with 60k samples that can be labeled for $35\$ * 50 + 10 * 25\$ = 2000\$$ with Google Cloud [1] and $0.04\$ * 60000 = 2400\$$ with Amazon Sagemaker [2]. Data collection here is practically free, since per category one can scrape the internet freely.

We model the defender cost as $c = n * (pl + cc)$ and the attacker cost as $c_a = n_a * (pl_a + cc_a)$, where $n$ is the number of points to annotate, $pl$ is the per-label cost and $cc$ is the data collection cost. We assume that the attacker wants to succeed in the attack and thus attacker's cost is less than that of a defender $n * (pl + cc) > n_a * (pl_a + cc_a)$.

First, we consider the attacker who extracts a model with DFME with 20 million queries. If we assume that the defender used Google Cloud to label their data, attacker's labels have to cost less than 0.00012\$:

$$20M * pl_a < 2,400\$ \rightarrow pl_a < 0.00012\$ \tag{1}$$

The queries in DFME are synthetic, therefore there is no data collection cost ($cc_a = 0$). Next, we consider a model extraction attack that only utilizes $5\% = 3000$ datapoints of prior knowledge, and no additional queries of any sort. Here, attacker breaks even at 0.8\$ per-label. Note that with 5% of data it is indeed possible to extract a model, while remaining well within a reasonable budget in the

---

[1] https://cloud.google.com/ai-platform/data-labeling/pricing
[2] https://aws.amazon.com/sagemaker/data-labeling/pricing/

current labeling market. At the same time, extraction with DFME stops being cost-effective when the attacker queries are even minimally priced by the defender.

This case study suggests the need for pricing of attacker queries and that model extraction might not be the most cost-effective solution, even for simple, commonly used benchmarks such as CIFAR10.

It is worth noting that this relationship does not always hold true in other cases. Medical data, for example, requires significant upfront equipment investment and needs prolonged interactions with data suppliers. This in turn means that the costs will be rather dominated by the data acquisition, as opposed to the labeling and such case will exhibit a different behaviour.

## B  A NOTE ON CURRENT LITERATURE

Active learning is a branch of semi-supervised learning which focuses on finding query efficient-training regimes, *i.e.* it explores methods that can learn a task with the least number of questions to the oracle. Chandrasekaran et al. formulated that 'the process of model extraction is very similar to active learning' and suggested that improvements in query synthesis active learning should directly translate to model extraction. This relationship works in both directions and implies that greater performance in model extraction directly translated to better active learning regimes.

Some of the current literature reports an ability to extract complex models with a handful of queries *e.g.* Tramèr et al. claims successful extraction of a Multilayer Perceptron with around a thousand queries or Zanella-Beguelin et al. extracts SST-2 BERT with around two thousand queries. This suggests that there exists (very) query-efficient training strategies, with oracles providing labels for otherwise unlabeled data. Yet, in practice, literature in active learning reports less impressive results in these settings. For example, sophisticated state-of-the-art regimes on CIFAR-10 report improvements up to 7% for 5% and up to 5% for 10% of dataset (Yoo & Kweon, 2019; Beck et al., 2021; Yi et al., 2022). Most importantly, the improvements are similarly dominated by the original data access.

Indeed, our paper questions the apparent *free lunch* reported in model extraction literature, and suggests that extraction reported is mostly an artifact of underlying data access, drastically overestimating potency of the attacks. To best understand the underlying attack performance it is imperative to consider the benefit of model extraction for no-data settings and cover them extensively, focusing on the low 0–5% regions.

## C  EFFECT OF SURROGATE DATASET OVER ATTACK PERFORMANCE

In Section 4.3 we investigated the performance gain of using additional queries drawn from a different data distribution. We evaluated this for a query budget of the size of the original training dataset. This budget represents the maximal query complexity that can still be considered successful. Here, we further extend the results presented in Figure 3 and evaluate the effect of the number of additional queries from various data sources.

Results are presented in Figure 6. We show that, in most cases, larger numbers of additional queries only have a limited effect, which is also very dependent on the quality of the queries and the level of prior knowledge. Adversaries with higher levels of prior knowledge are less affected by the additional queries. The DFME queries, while being out-of-distribution, do perform very well, but at the cost of prohibitively high query budgets.

## D  ATTACK CONVERGENCE RATE COMPARISON

Following the discussion in Section 4.2, we further investigate the actual benefit of attacking the model rather than training from scratch, from the computational perspective. We examine the convergence rate of the attack accuracy. Figure 7 presents a comparison between the convergence rate of the three cases - an attacker that uses the victim's soft labels (*i.e.* full probability vectors), an attacker that has a label-only access to the victim model, and an attacker that uses the real (ground truth) labels. We show that the convergence rate is similar across the three attackers, *i.e.* the attacker does not "learn faster" by querying the victim model. We show that in cases where the attacker has limited prior

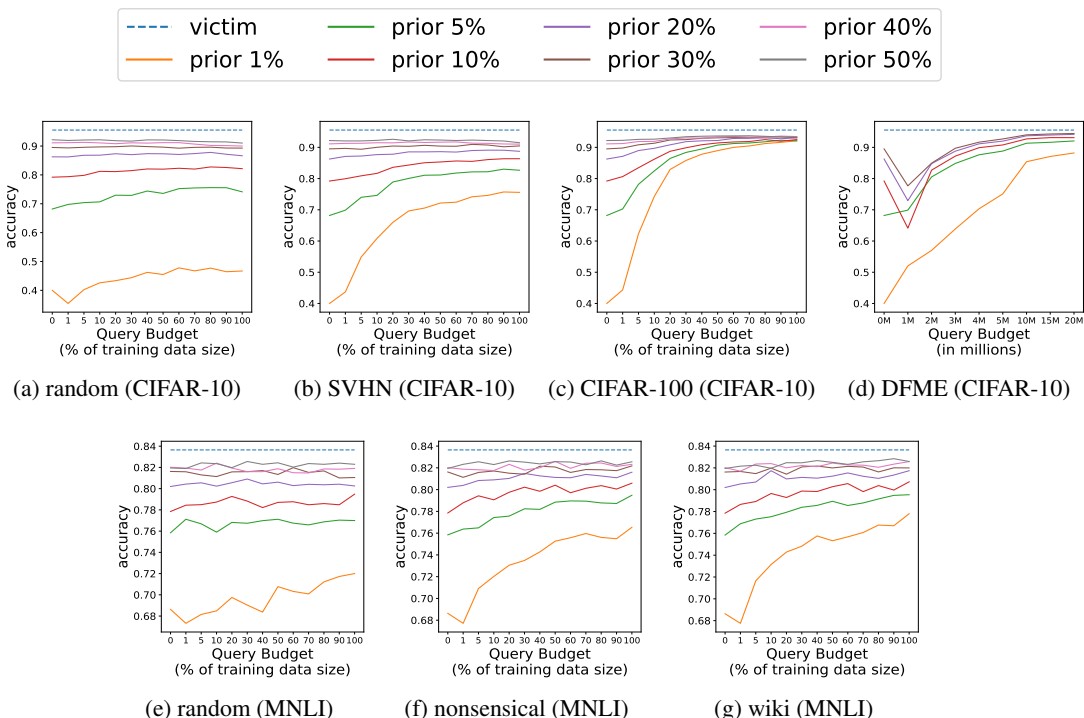

Figure 6: Evaluating the effect of adding different amounts of additional queries for a given level of attacker prior knowledge. Plots (a)-(d) present attacks over the CIFAR-10 victim model and (e)-(f) are for the MNLI victim model. In all plots, a query budget of 0 represents the baseline attack accuracy, as presented in fig. Figure 2. Results show that adding larger amounts of additional queries has a limited effect, which is dependent on the query quality and amount of prior knowledge.

knowledge over the data distribution, *e.g.* in the $5\%$ case, the attacker does get some benefit from using the victim's soft labels; however, this benefit disappears with increased prior knowledge.

# E    SAMPLING COMPLEXITY INTUITION

In Section 5.1 we discuss the intuition behind the complexity of sampling IND queries with or without prior knowledge over the distribution. Here, we provide a toy example of this complexity for different levels of prior knowledge, modeled by the overlap between the IND and the attacker's query distribution. We then extend intuition for our considered models, and estimate the complexity of sampling IND in this setting.

## E.1    SAMPLER BIAS TOY EXAMPLE

We explore the relation between a prior knowledge level and the attacker's probability of successfully sampling from the useful domain. We denote the sampler from the useful domain, *i.e.* the in-distribution domain, as $V_s \sim \mathcal{N}(\mu_v, \sigma_v)$, and the attacker's sampler, *i.e.* the distribution from which queries are drawn, as $A_s \sim \mathcal{N}(\mu_a, \sigma_a)$. We model the prior knowledge level as the Wasserstein distance between both distributions.

Figure 8 plots the probability of sampling from the "informative" overlap region as a function of Wasserstein distance between $V_s$ and $A_s$ when $\mu$ and $\sigma$ are sampled uniformly. Small differences in sampling distributions, *i.e.* less prior knowledge, result in a significant reduction in in-distribution sampling probability. This, in turn, results in *wasted queries*, as sampling outside of the overlap is not informative, and in *reduced model capacity*, as the attacker "wastes" capacity on learning the irrelevant OOD region. This holds even in cases where distributions overlap significantly. Note that in

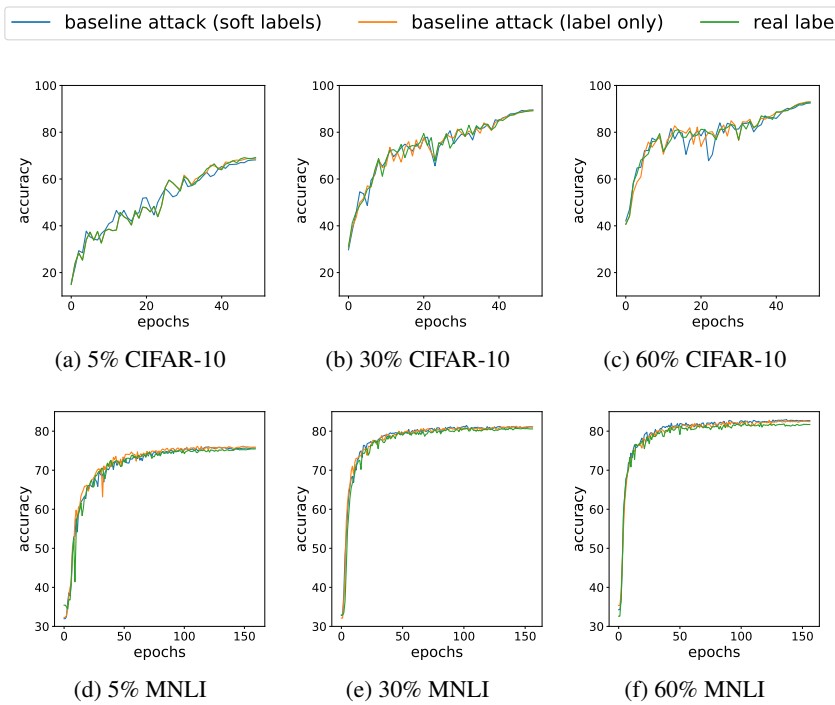

Figure 7: Comparison between the convergence rate of an attacker that uses the victim's full probability vector output (soft labels), an attacker that utilizes a label-only access to the victim model, and an attacker that uses the real ground truth labels. In all cases the attacker has access to $30\%$ of the true training samples. The attacker does not "learn faster" by attacking the victim model, and only benefits from the victim model when it has little prior knowledge over the true data distribution.

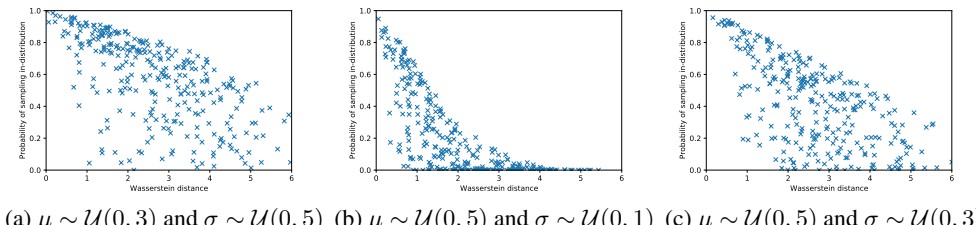

Figure 8: Gaussian overlap as a function of Wasserstein distance.

practice, with more dimensions, the volume would overlap less, and the useful sampling probability would be even further reduced.

### E.2 SAMPLING COMPLEXITY IN REAL MODELS

In the previous section, we explore the sampling complexity of an attacker in a toy setting. In this section, we extend this notion to a more realistic scenario of estimating the sampling complexity for our real victim models. For this, we attempt to estimate the in-distribution volume and the complexity of sampling within it by measuring the number of random queries in this volume. Similarly to our OOD component, described in detail in Section 5.3.1 and Appendix F, we define a query as in-distribution, *i.e.* inside the volume, by observing the model's confidence in predicting this query. High confidence queries, above some predefined confidence threshold, are considered to be in-distribution. Therefore, for the task of volume estimation, we measure the percentage of random queries that are above this threshold. In this section, we use a threshold value of 90.

In Figure 9 (a) we present our estimation in both victim models. As can be seen, only $59\%$ of the random queries are sampled from within the volume in the case of the CIFAR-10 model, and only $45\%$ in the case of the MNLI model. We additionally compare this to the sampling complexity when using real in-distribution data, by measuring the percentage of samples from the test set that are predicted by the model with a high confidence. In this case, where we have significant prior knowledge over the distribution, the sampling complexity is drastically decreased.

As our estimation dependents on the model confidence, it is also interesting to observe how the complexity changes when the model becomes less confident. For this reason, in Figure 9 (b) we perform the same evaluation, however, we decrease the victim model's confidence by increasing its softmax temperature from 1 to 2. This change results in a less confident model in general and, as evident from the results, it is nearly impossible to sample from within the distribution without any prior knowledge: only $0.00001\%$ for CIFAR10 and $0.01$ for MNLI.

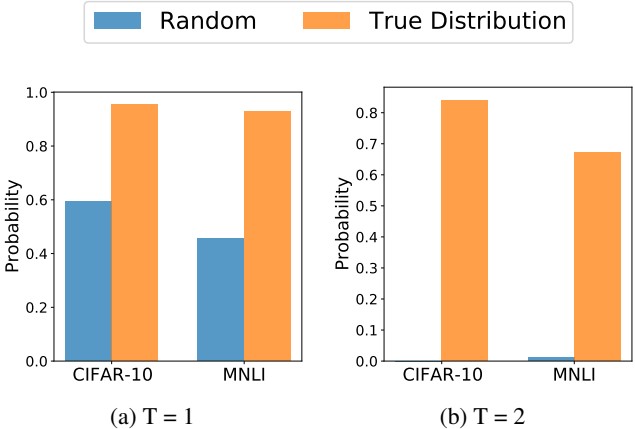

(a) T = 1                  (b) T = 2

Figure 9: Estimating the sampling complexity of our real victim models by measuring the percentage of random queries that are predicted by the model with high confidence, *e.g.* above $90\%$. This is in comparison to the complexity when using queries from the true distribution, sampled from the test set. We evaluate this for both the original model, with a softmax temperature T=1 (a), and for a less confident model, with a high softmax temperature T=2 (b).

## F    OOD FULL IMPLEMENTATION DETAILS

In this section, we elaborate on the implementation details behind our OOD component, described shortly in Section 5.3.1. As described before, given the original victim model $\mathcal{V}_o$, we create a hybrid victim model $\mathcal{V}_h$ by combining the original victim model $\mathcal{V}_o$ with an additional module with different, or additional, decision boundaries $\mathcal{V}_f$. For each query $x$, we apply a decision rule $R$ over $x$, to determine which of the two modules, $\mathcal{V}_o$ or $\mathcal{V}_f$, should be used for predicting this query.

The decision rule $R$ is implemented by applying a threshold value $\tau$ over the prediction confidence of $\mathcal{V}_o$. If a query $x$ has a low prediction confidence, we define it as an OOD query and return the prediction $\mathcal{V}_f(x)$. Otherwise, we define it as IND and return $\mathcal{V}_o(x)$. We calibrate $R$ by increasing the model's Softmax temperature to 2, as discussed in more detail in Appendix F.2. As mentioned in Section 5.3.1, thresholding the prediction confidence serves as a naive approximation for OOD detection, and using more sophisticated OOD detection methods will only increase the effect of our OOD component. As such, the exploration of different decision rules is out of scope for this work.

As for the fake model $\mathcal{V}_f$, we implement it by fitting a Gaussian Mixture Model (GMM) for each class of the training data. For each class $c \in \{1, 2, ..., C\}$, where $C$ is the number of classes, we sample $S = 5000$ training data points labeled as class $c$, *i.e.* $\{(x_1, c), (x_2, c), ..., (x_S, c)\}$. We then use the victim model $\mathcal{V}_o$ to compute the predictions of this set, *i.e.* the logits. We fit a GMM to this set of logits $\{\tilde{y}_1, \tilde{y}_2, ..., \tilde{y}_S\}$, where $\tilde{y}_i = \mathcal{V}_o(x_i)$.

For each class, we create $m = 5$ anchor points $\mathcal{A}_c^1, \dots, \mathcal{A}_c^m$ that will be used to "assign" queries for this class. For this, we first compute the feature representations of the data samples used for fitting

the class GMM,*i.e.* $\{(x_1, c), (x_2, c), \ldots, (x_S, c)\}$, using some feature extractor $\phi$. Then, we cluster these feature vectors $\{\phi(x_1), \ldots, \phi(x_S)\}$ into $m$ clusters using the Kmeans clustering algorithm. We define the anchor points to be the centroids of each cluster. For vision tasks, we use a pre-trained ResNet34 for the feature extractor $\phi$; for NLP tasks we use a pre-trained BERT-base model.

At last, we sample $m$ permutations $\pi_j : C \to C$, for $j \in [m]$. The permutations would ensure that no query would be predicted using its "real" assigned class, therefore avoiding tail-of-distribution samples classified as OOD samples (*i.e.* false-positives samples), to be correctly labeled and leak information about true IND behaviour.

For a given query sample $x$, we compute its feature representation $\phi(x)$ and find the nearest anchor point $\mathcal{A}_i^j$, in terms of the $L_2$ distance between $\phi(x)$ and all $C \times m$ anchor points. The anchor point $\mathcal{A}_i^j$ represent the $j^{th}$ anchor point of the $i^{th}$ class. We then permute the assigned class $i$ using the $j^{th}$ permutation,*i.e.* $i' = \pi_j[i]$, and sample a "fake" logit $\tilde{y}$ from the GMM we fitted earlier to class $i'$. This value is returned as the complex model prediction, $\mathcal{V}_f(x) = \tilde{y}$.

The construction of our OOD component requires the evaluation of a subset of the training data used for fitting the GMMs. The evaluation, fitting, and clustering process are done only once; hence, it is relatively computationally inexpensive. During inference, each query directed to the OOD component adds some computational cost of computing its feature representation, and performing a nearest neighbor search. However, this is only true for OOD samples and false-positive IND samples. Most legitimate users' queries, *i.e.* the true-positive queries, are completely unaffected by our OOD component.

## F.1 EFFECT OF ANCHOR POINTS AND PERMUTATIONS

The design of our proposed OOD module $\mathcal{V}_f$ satisfies two main objectives. The first, and most important, is to avoid leaking real IND behaviour by the predictions made by $\mathcal{V}_f$. The second, was to lure the attacker to waste effort and capacity in learning fake and more complex decision boundaries, and therefore reducing further it's performance over the real IND decision boundaries.

To obtain the first goal, we introduced the permutation mechanism. As mentioned before, some tail-of-distribution samples are falsely classified as OOD samples, *e.g.*false-positives. As these samples share the same distribution as the anchor points, when searching for the nearest neighbor anchor point, they would be correctly assigned to their real class. When no permutation is applied, this will result in a prediction that resembles the original prediction given by $\mathcal{V}_o$, and therefore leaks information. To avoid this, the matching anchor point should direct the sample to a different class by using some class-level permutation.

To obtain the second goal, we introduce multiple new decision spaces by using multiple permutations per class. As a result, two samples that were initially assigned to one class can now be directed to two different classes, placing a new decision boundary between them. For this, we use multiple ($m = 5$) anchor points per class, and each anchor point is coupled with a different permutation. Therefore, two samples $x_1$ and $x_2$ that were assigned to two anchor points related to class $i$: $\mathcal{A}_i^1$ and $\mathcal{A}_i^2$, will now we reassigned to two different classes. $x_1$ would be predicted using the GMM fitted for class $j = \pi_1(i)$ and $x_2$ would be predicted using the GMM fitted for class $k = \pi_2(i)$.

To better demonstrate the effect of using multiple permutations and anchor points, we provide an ablation study in Figure 10. We compare the performance of our OOD module in 4 different settings: (i) using one anchor point per class and no permutations (ii) using one anchor point per class with class-level permutation (iii) using 5 anchor points per class with the same permutation shared between all anchor points (iv) our proposed method - 5 anchor points per class with 5 different permutations. These would be denoted as "1 anchor, no perm", "1 anchor, 1 perm", "5 anchors, 1 perm", "5 anchors, 5 perms", respectively. The biggest effect can be attributed to the simple addition of permutations; however, it can be further emphasized by incorporating the additional anchor points and multiple permutations.

## F.2 SOFTMAX TEMPERATURE INFLUENCE

In order to calibrate $R$ to better distinguish between IND and OOD samples, we increase the model's Softmax temperature to 2. By doing so, we force the model to be less confident, which results in

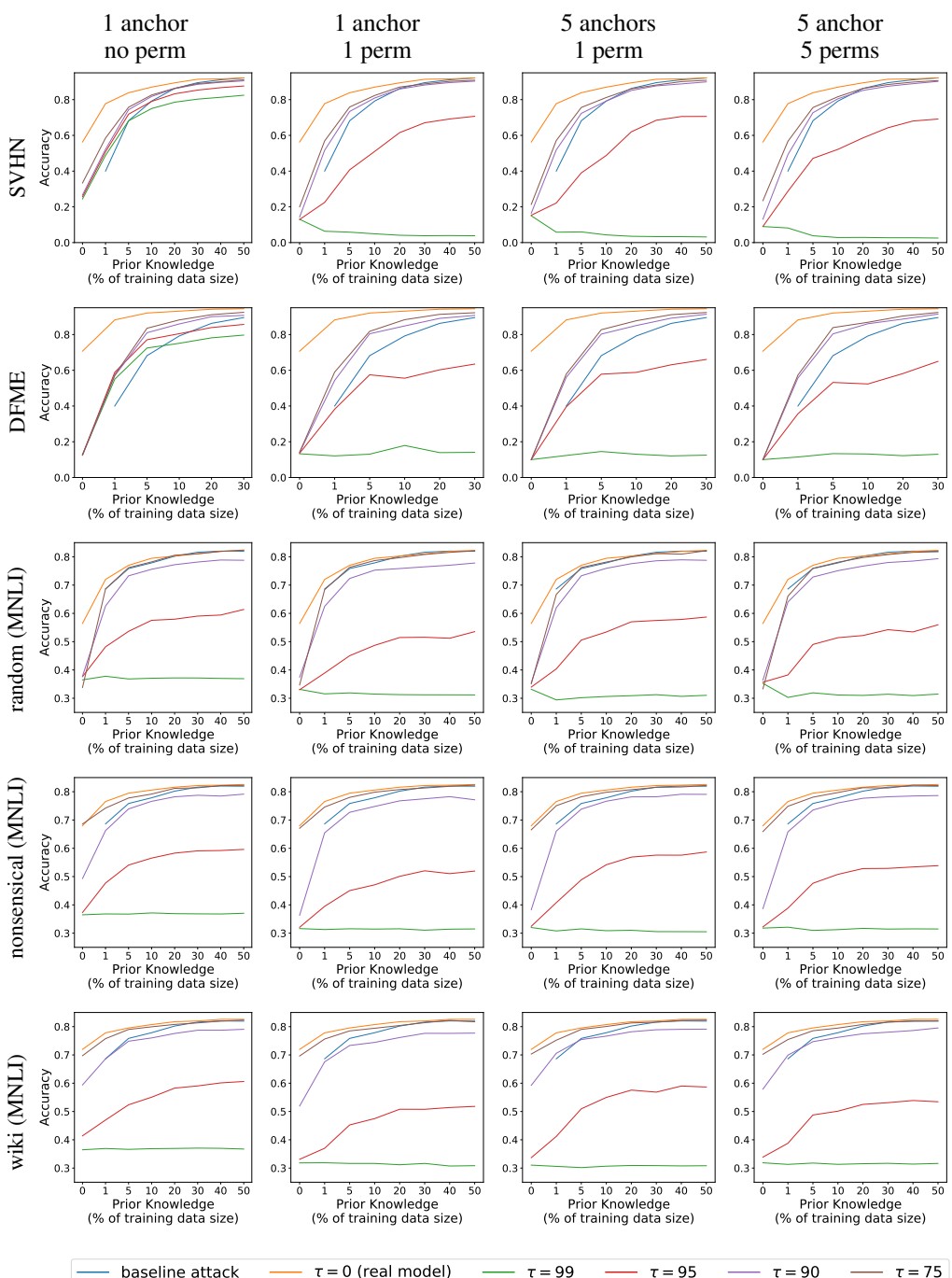

Figure 10: An ablation study of the use of multiple anchor points and permutations in the instrumentation of the OOD module. Comparing between 4 settings shows that indeed our proposed method ("5 anchors, 5 perms") results with the biggest degradation of the attacker's performance even when given access to higher levels of prior knowledge.

more queries being predicted by $\mathcal{V}_f$ and not $\mathcal{V}_o$. As shown in Figure 9, a temperature value of 2 indeed detects most of the OOD queries. In Figure 11 we demonstrate the effect of the temperature value over the behaviour of our OOD component, and show that a better calibrated OOD component indeed further lowers the benefit the attacker gains by utilizing OOD queries.

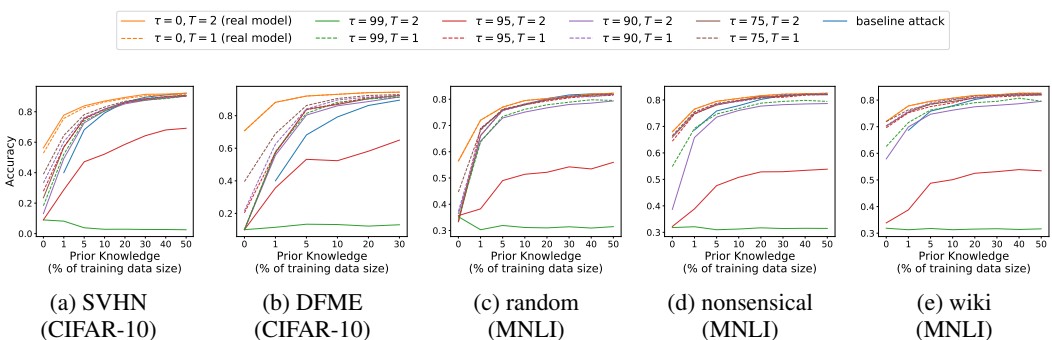

|           |           |           |           |           |
| :-------: | :-------: | :-------: | :-------: | :-------: |
| (a) SVHN  | (b) DFME  | (c) random | (d) nonsensical | (e) wiki |
| (CIFAR-10) | (CIFAR-10) | (MNLI)   | (MNLI)    | (MNLI)    |

Figure 11: As our OOD component is distinguishing between in and out of distribution queries based on the model confidence, we evaluate the effect of decreasing the model's confidence by increasing the softmax temperature T from 1 to 2. Results shown in this plot show that this indeed even further lowers the benefit the attacker gains by utilizing OOD queries.

It is important to note that this also increases the FPR of the IND samples, *i.e.* the percentage of IND samples predicted by $\mathcal{V}_f$ instead of $\mathcal{V}_o$. A higher FPR results in a decline in model test accuracy. However, as shown in Table 1, and described in more detail in Appendix F.3, this decrease is not linearly dependent on the exact FPR.

### F.3 Impact of $\tau$

The value of the confidence threshold $\tau$ determines the utility – extraction difficulty trade-off. In Section 5.3.2 we evaluate our OOD component on different values of $\tau$, and observe the effect each has over the attack performance. In order to verify that our OOD component is not trivially rejecting all inputs, we additionally measure the false-positive rate (FPR) for the different threshold values. In Table 1, we detail the relation between the threshold value, the FPR, and the effect on the victim's test accuracy. For lower $\tau$ values, most queries are predicted using the original $\mathcal{V}_o$, *i.e.* the victim model's utility is barely harmed by adding the OOD component since most queries are answered by the real model. In this case, the attacker's performance is barely affected, as it can still get informative responses by issuing OOD queries. As $\tau$ increases, more queries– including some IND ones– are predicted by the "fake" model $\mathcal{V}_f$. This makes it more difficult for an attacker to infer which decision boundaries are IND. We discussed the underlying complexity in detail in Sections 5.1 and 5.2. As can be seen, the attack accuracy dramatically decreased, which verifies that the OOD behaviour leaks almost no information about the IND behaviour. To further demonstrate this, we present in Figure 12 a comparison between the labels predicted by $\mathcal{V}_o$ and those predicted by $\mathcal{V}_f$ for actual OOD queries (*i.e.* true positives) as well as tail-of-distribution IND samples that were detected as OOD (*i.e.* false positives). We show this comparison for an attacker that utilizes additional SVHN queries, 30% prior knowledge, and for a threshold of $\tau = 95$. It is clear that, although the "fake" labels are heavily biased towards one class for the additional queries, in both cases, they are uncorrelated with victim models' predicted labels.

### F.4 On learning of fake boundaries

Although modern learning is an inherently stochastic process, learning with standard tools such as SGD has a bias towards structured solutions (Soudry et al., 2018; Mousavi-Hosseini et al., 2022). For the same model parameter budget, SGD learns the smoothest boundaries first and only gets to the other boundaries if the capacity permits (Ben Arous et al., 2021). This has real practical implications on the fake regions that we add to the model. Namely, we can not add decision boundaries that are more complex than the real task, since SGD learns to ignore them in light of the real decisions, limiting the extent to which we can add arbitrary complexity into the models.

In Section 3 we have discussed the query budget and model capacity that the attacker must spend if it can not distinguish between the task-related (IND) and unrelated (OOD) queries. To verify, we explicitly check that the attacker indeed learned both the task related and unrelated knowledge, and did not "ignore" the predictions made by $\mathcal{V}_f$ due to the SGD bias described above.

| $\tau$ | CIFAR-10 | | | | MNLI | | | |
| | Temp. 1 | | Temp. 2 | | Temp. 1 | | Temp. 2 | |
| | FPR | Acc | FPR | Acc | FPR | Acc | FPR | Acc |
|---|---|---|---|---|---|---|---|---|
| 0 | 0 | 95.5 | 0 | 95.5 | 0 | 83.6 | 0 | 83.6 |
| 75 | 2.5 | 94.5 | 7.4 | 91.4 | 3.5 | 83.2 | 9.1 | 81.9 |
| 90 | 4.7 | 93.2 | 15.9 | 84.1 | 7.1 | 82.4 | 32.8 | 72.8 |
| 95 | 6.5 | 92.0 | 40.5 | 60.4 | 10.4 | 81.5 | 71.1 | 49.6 |
| 99 | 10.6 | 88.92 | 99.7 | 2.32 | 27.7 | 75.2 | 100 | 32.1 |

Table 1: The threshold value $\tau$ determines which samples are marked as in-distribution, predicted by the original model $\mathcal{V}_o$, and which are treated as OOD and predicted by $\mathcal{V}_f$. The higher the threshold, the higher the false positive rate is. This, in turn, has a negative effect over the teacher test accuracy. This also implies that higher threshold values result in a higher true positive rate. The Softmax temperature (denoted as "Temp." in the table) also affects the FPR and can be used to better calibrate the OOD detection component.

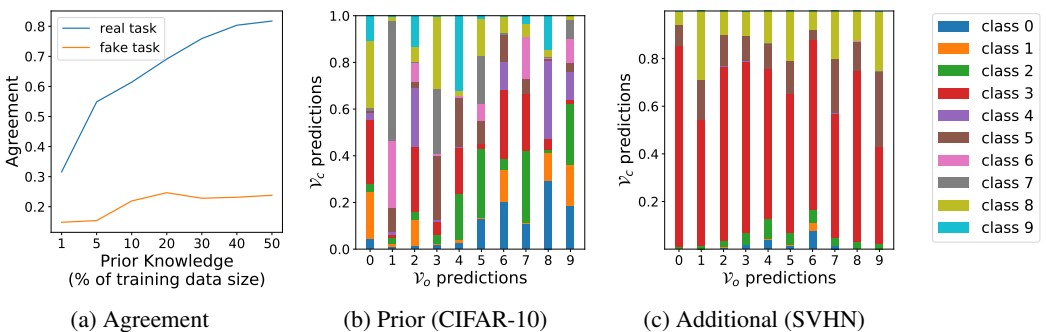

(a) Agreement   (b) Prior (CIFAR-10)   (c) Additional (SVHN)

Figure 12: (a) The agreement between the attacker model and the victim model, for the CIFAR-10 task and an attacker utilizing SVHN additional queries. The agreement is separated into the real task ($\mathcal{V}_o$) and the fake task ($\mathcal{V}_f$). We can see that fake task agreement is higher than random guessing (10%), which implies that the attacker was able to learn the fake, irrelevant task, and waste some capacity.

(b)-(c): Comparison between the labels predicted by $\mathcal{V}_o$ (x-axis) and the labels predicted by $\mathcal{V}_f$ (y-axis), for an attacker that utilizes additional SVHN queries, 30% prior knowledge, and for a threshold of $\tau = 95$. Each bar $i$ demonstrates the distribution of the "new" fake labels that would have been predicted by $\mathcal{V}_o$ as class $i$. The y-axis is normalized to show the percentage of samples from each class. Although the fake models' predictions are biased towards one class for the additional queries, in both cases they are un-correlated with the victim models' predictions, and therefore present new decision boundaries for the OOD queries.

For this reason, we investigate the agreement between the attacker model and both $\mathcal{V}_o$ and $\mathcal{V}_f$, for the SVHN additional queries case. We separate the agreement of the samples predicted by $\mathcal{V}_o$ and the samples predicted by $\mathcal{V}_f$. Figure 12 demonstrates that the attacker model agreement with $\mathcal{V}_f$ is higher than chance level, which is 10% in this case (for the 10 CIFAR-10 classes). This proves that the attacker learned both the real and the fake task and, as such, wasted capacity on learning irrelevant decision boundaries.

## G KNOCKOFF NETS EVALUATION

In addition to the CIFAR-10 and MNLI datasets, we evaluate our main experiments on the Indoor67 (Quattoni & Torralba, 2009), CUBS200 (Wah et al., 2011) and Caltech256 (Griffin et al., 2007) datasets. For this, we follow the setting and training details considered by the Knockoff Nets

attack (Orekondy et al., 2019a), and use the pre-trained victim models provided by the authors. Orekondy et al. provides two strategies for sampling queries (i) random - in which the queries are sampled uniformly at random from some query distribution (ii) adaptive - in which the queries are sampled according to a learned policy $\pi$. We note that there is no official implementation for the adaptive strategy, and thus we reimplement the method with the details provided by Orekondy et al.. We simplify the hierarchy to be one-level deep, and omit the coarse-to-fine label hierarchy used to supplement the policy. We observe similar performance to the original results in (Orekondy et al., 2019a).

For the attacker model, we use a ImageNet pretrained ResNet-34 architecture. We refer the reader to Orekondy et al. for full training details.

### G.1 BASELINE ATTACK

We begin by evaluating our baseline attacker, following the setting described in Section 4.2. We assume the attacker has access to either a randomly or adaptively sampled subset of the victim's true training dataset. The results, presented in Figure 13, align with our previous findings and show that in most cases, an attacker with access to more than $50\%$ can almost fully extract the model. These results align with the results presented in Orekondy et al.'s Figure 5.

As in Section 4.2, we compare the attack performance using soft-label access to the victim model to that of an attacker with access to the real (ground truth) labels or label-only access to the victim model. This comparison provides the same evidence as described in Section 4.2. It demonstrates that attacking the victim model is merely using the victim model as a labeling oracle in the absence of access to the real labels, and the attacker does not gain much additional benefit. This phenomena was also observed by Orekondy et al..

### G.2 SURROGATE DATASET

We investigate the attack performance gain from the use of additional queries, sampled from a data distribution that differs from the train distribution. Following Orekondy et al. we use ImageNet (Deng et al., 2009) as the surrogate dataset, from which we sample randomly or using the adaptive strategy. We follow the setting described in Section 4.3, and fix the query budget to be the size of the original training dataset. The results, presented in Figure 14 align with the results in Figure 3, as they show that as the attacker has more prior knowledge, it benefits less from utilizing additional queries.

We do observe that in the lower prior knowledge settings, the ImageNet additional queries do provide a significant improvement, especially for the Indoor67 and Caltech256 datasets. We hypothesize that this is due to a high similarity between the distributions. The results clearly demonstrate that the attacker's performance is dominated by the access to IND data. The effect of the query selection strategy is inferior to that of data access, which aligns with the main findings of our paper.

### G.3 LIMITING OUT-OF-DISTRIBUTION EFFECTIVENESS

We explore the reliance of the Knockoff Nets attacker on the implicit assumption that the IND decision boundaries can be inferred from the OOD ones. For this, we limit the informativeness of the OOD ImageNet queries by incorporating our OOD detection component. We follow the same setting as in Section 5.3.2 and evaluate the effect using different threshold values $\tau$. We set the model's softmax temperature to 2, and use $\|D_{train}\|$ additional queries.

Figure 15 demonstrates that limiting the informativeness of OOD queries indeed decreases the benefit the attacker gained by utilizing additional ImageNet queries, even to the point of performing worse than the baseline in the higher prior knowledge settings.

## H NLP DISTRIBUTION SIMILARITY

The results presented in Figure 4 show that the effect of the OOD component is weaker for some of the settings in our NLP task, specifically the nonsensical and wiki queries. This might seem confusing. The wiki queries are broadly equivalent to the surrogate dataset queries that are commonly used in the vision domain, however they exhibit a different response to our OOD component. As both

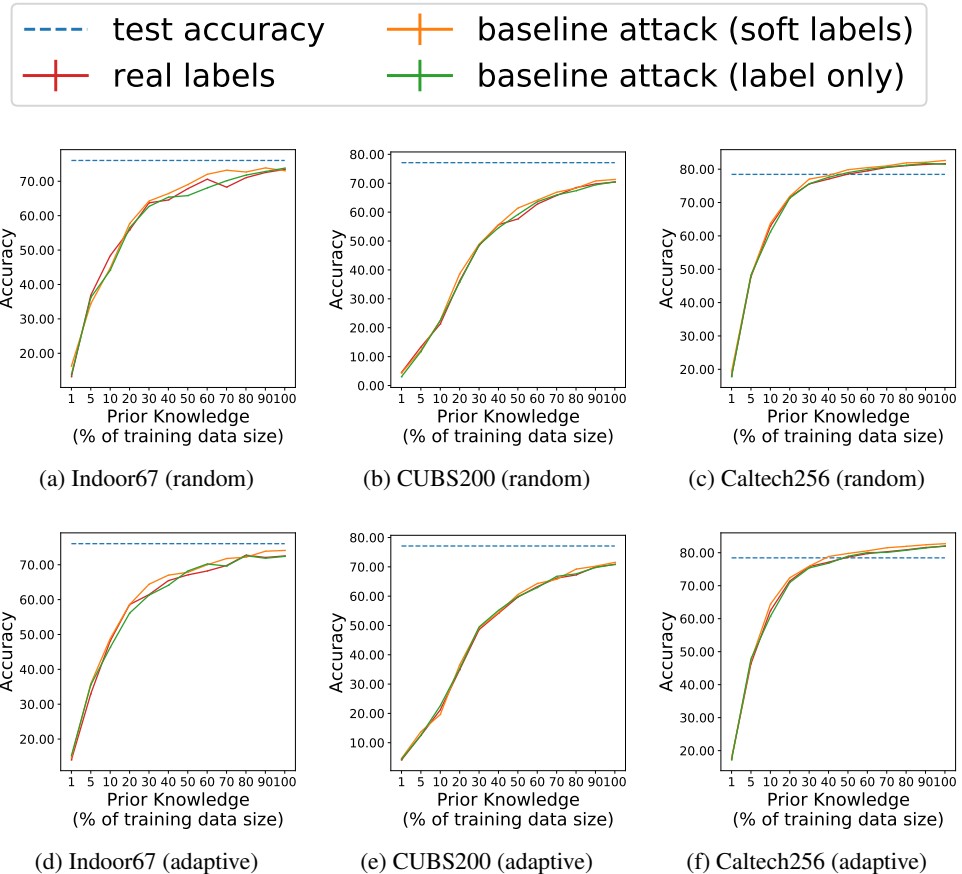

Figure 13: Empirical evaluation of the risk posed by an attacker with some prior knowledge over the true data distribution. The prior knowledge is expressed as access to a percentage of the true training set. In most cases, an attacker with more then $50\%$ access can nearly fully extract the model, however, in this case, the extraction is not more efficient than training from scratch. The attacker does not gain much by querying the victim model versus using the real labels, which also seems to be equivalent to a label-only access to the victim model. This shows that, other than performing as a labeling oracle, the extraction is meaningless in this case.

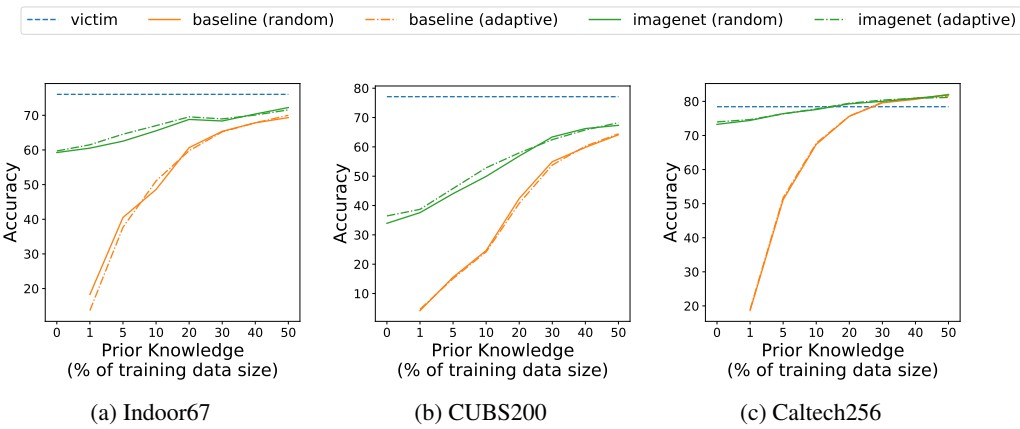

(a) Indoor67      (b) CUBS200      (c) Caltech256

Figure 14: We evaluate the effect of augmenting the attacker's queries with additional queries sampled from the ImageNet dataset, in comparison to our baseline attacker, which only use its prior knowledge. We fix the query budget to be the size of the original training set to provide a fair comparison between the different attackers. It can be seen that, as the attacker has more prior knowledge over the true distribution, it does not gain much benefit by augmenting the query set. Since the ImageNet dataset does share some distributional similarity with the true training data distributions, we observe a significant improvement from utilizing it in the lower prior knowledge settings.

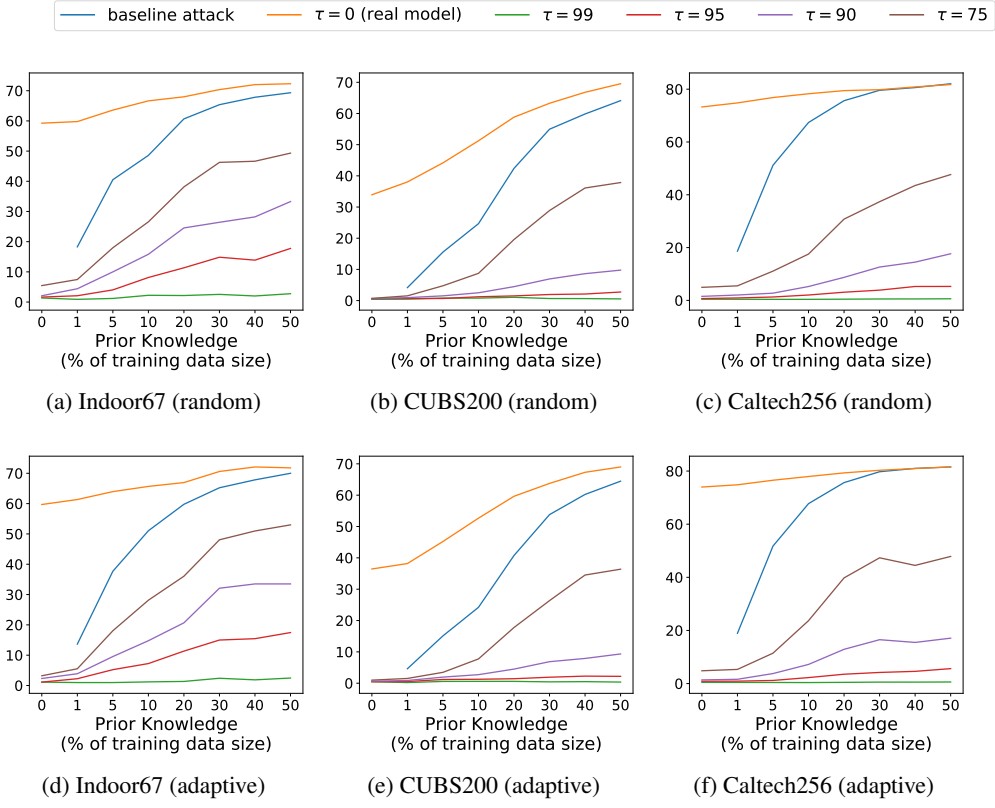

(a) Indoor67 (random)      (b) CUBS200 (random)      (c) Caltech256 (random)

(d) Indoor67 (adaptive)      (e) CUBS200 (adaptive)      (f) Caltech256 (adaptive)

Figure 15: The effect of applying our OOD component with different values of $\tau$ against an attacker that utilizes $\|D_{train}\|$ additional ImageNet queries. When comparing the results to the original setting (real model), where the OOD region is unmodified, we can see a clear decrease in the attack accuracy.

random images and random nonsensical sentences have no real meaning, one might expect similar behaviour between them. In the past, nonsensical queries were even used as a benchmark under the name of random queries *e.g.* by Truong et al..

This difference between the vision and NLP domains is due to the similarity between the true data distribution and the aforementioned query distribution. Although the queries are drawn from either a random nonsensical distribution in the *nonsensical* case, or a from a different corpus in the *wiki* case, we observe that many common words are shared between all three distributions. This results in similar model behaviour between the OOD queries and the true data. It can explain the success of the queries in this domain, which comes in significant contrast to the lack of success of random or some surrogate queries in the vision domain, where the input space is a continuous pixel space rather than a (relatively small) discrete dictionary of words. The similarity between distributions results in high confidence values for both in-distribution and out-of-distribution queries, making it difficult to separate them and apply our OOD component. We show the confidence distribution for each query type in Figure 16. In the case of the random queries, where each letter is sampled, and the sentences are not composed of real words, we can observe a significantly lower gain from the additional queries, which is more in line with the findings from the vision domain.

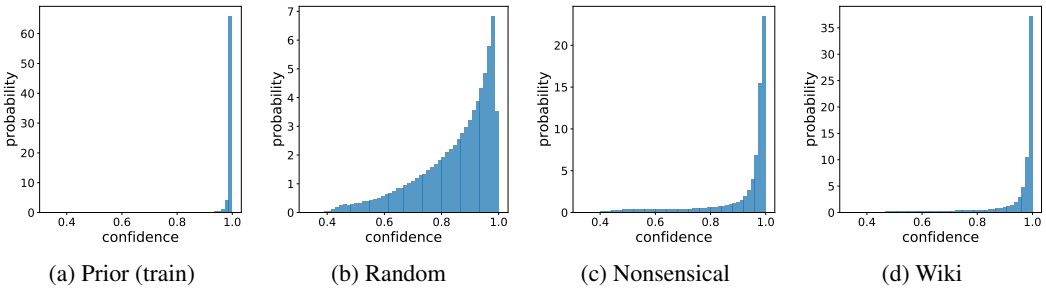

|(a) Prior (train)|(b) Random|(c) Nonsensical|(d) Wiki|

Figure 16: We compare the confidence values of the victim model over the true data distribution versus the different types of out-of-distribution queries. Results show that the model tends to be highly confident for both in-distribution as well as for some of the out-of-distribution queries, making it harder for the OOD component to make any impact for lower threshold values. This result can explain the surprising success of using random nonsensical queries in the language domain versus the poor results of using random queries in the vision domain.

## I   TASK AND MODEL COMPLEXITY

In this section, we investigate the relationship between task and model complexity, as well as the query complexity of model extraction attacks. In Figure 17 we observe that for simpler tasks, such as classification over the SVHN and SST-2 datasets, even an adversary with little prior knowledge can successfully extract the model, *e.g.* $5\%$ data access. This is in contrast to the results we demonstrated for the CIFAR-10 and MNLI datasets, which are significantly harder tasks to learn.

We additionally investigate the effect of the size of the victim model architecture. In addition to the ResNet-34-8x CIFAR-10 victim model evaluated so far, we trained two CIFAR-10 victim models: a larger ResNet-50-8x and a smaller ResNet-18-8x. Note that we trained these models ourselves, while for the ResNet-34-8x architecture, we used the pre-trained model by Truong et al.. The results, presented in Figure 18, show that this has minimal impact on the attack success. Lack of differences here can be explained by the fact that all three models share similar accuracy - $94.22\%$ for ResNet-18-8x, $95.54\%$ for ResNet-34-8x, and $93.72\%$ for ResNet-50-8x.

As previously noted, the models in practice mainly serve as labeling oracles, meaning the difference in the attack performance is connected to the model accuracy. Having said that, it is worth mentioning that we have not performed a thorough hyperparameter search in the training of the models. This should not affect the validity of the results and should only cause a slightly lower accuracy.

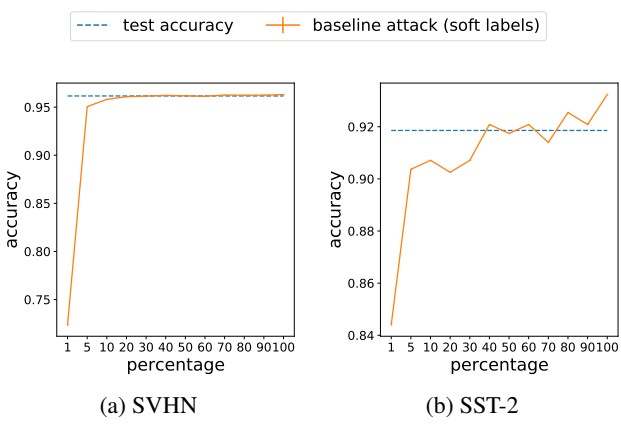

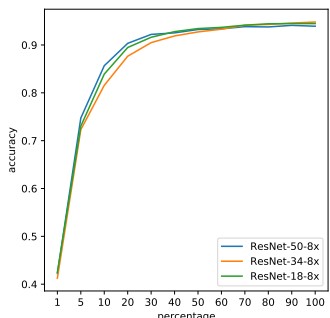

<table>
</table>

<p style="text-align:center">(a) SVHN        (b) SST-2</p>

Figure 17: The baseline attacker is able to successfully extract some victim models very easily, with as little as $5\%$ of the training data. We hypothesize that this is due to the inherent linearity of these datasets.

Figure 18: We investigate whether the victim model size has any impact on the attack success. We observe no impact when increasing (ResNet-50-8x) or decreasing (ResNet-18-8x) the model size. All models are incredibly over-parameterized and, therefore, the differences between the models are insignificant in terms of the attack complexity.

## J    TOWARDS EVALUATION OF MODEL EXTRACTION ATTACKS

In this paper, we investigated the performance of model extraction attacks in a variety of data access settings. We discovered that for the attacks to be efficient, the adversary either has to have access to in-distribution data or perform a very large number of queries. In cases where data access is restricted, in line with active learning, as noted in Appendix B, the adversary can only get slight benefits. Furthermore, the performance itself is dominated by the underlying data access, meaning that collecting more clean data gives strictly more performance and removes the need to query the victim altogether.

This raises the question of how one should evaluate model extraction attacks. First, the attack should be compared to an appropriate baseline, namely the performance of a model with just base data access evaluated against model training with additional queries. Second, since most model extraction benefits are observed at low data-access regimes, they should be evaluated exhaustively, empirically covering the range of 0–5% of the dataset size. Finally, the leakage out of distribution should be accounted for. The model is not expected to produce a meaningful decision for an unrelated task, so why would it leak information about model internals? As such, evaluation of model extraction attacks should include instrumentation with an OOD component and demonstrate the benefits of an attacker querying with it in place.

