# OpenReview forum: "Beyond Labeling Oracles: What does it mean to steal ML models?"
_ICLR.cc/2024/Conference — Submitted to ICLR 2024_

### Official Review · Reviewer_ersf · 2023-10-29

**Soundness:** 2 fair
**Presentation:** 3 good
**Contribution:** 3 good
**Rating:** 5
**Confidence:** 3

**Summary:**

The paper examines model extraction attacks, which aim to steal machine learning models via query access. It argues that the practical threat of these attacks is often overstated, as attackers frequently require prior access to in-distribution data. When such data is available, it is often more cost-effective to train a model from scratch, reducing the incentive for model extraction. The paper introduces a new benchmark to assess attacker knowledge and control the impact of out-of-distribution queries.

**Strengths:**

- The paper highlights the potential overestimation of the practical threat associated with such attacks.
- Use of illustrative examples to enhance understanding.
- Discuss the impact of surrogate datasets on model extraction performance.

**Weaknesses:**

- The paper's linear classification example may not align perfectly with the concept of accuracy.
- The rationale behind the mixed use of original and surrogate datasets requires further clarification.

**Questions:**

- The authors utilize linear classification as an illustrative example to convey the idea that in model extraction (ME) attacks, adversaries require either prior knowledge about the data distribution or the ability to make a substantial number of queries to the target model. However, in my interpretation, this example primarily underscores that these conditions are crucial for preserving high fidelity rather than accuracy. The focus of the adversary in this case centers on replicating the decision boundary. While accuracy and fidelity share similarities in certain contexts, distinct decision boundaries may result in comparable performance but differ in fidelity. Consequently, it would be more appropriate for the authors to place greater emphasis on the concept of fidelity in their paper.

- In Section 4.3, the authors discuss the ME performance gains from the use of surrogate datasets. However, they don't just use surrogate data alone; instead, they mix part of the original dataset with the surrogate data. It would be beneficial to understand the rationale behind this mixed approach.

---

> ### Author Response · Authors · 2023-11-20
>
> We thank the reviewer for the careful review.
> ```
> The authors utilize linear classification as an illustrative example to convey the idea that in model extraction (ME) attacks, adversaries require either prior knowledge about the data distribution or the ability to make a substantial number of queries to the target model. However, in my interpretation, this example primarily underscores that these conditions are crucial for preserving high fidelity rather than accuracy. The focus of the adversary in this case centers on replicating the decision boundary. While accuracy and fidelity share similarities in certain contexts, distinct decision boundaries may result in comparable performance but differ in fidelity. Consequently, it would be more appropriate for the authors to place greater emphasis on the concept of fidelity in their paper.
> ```
> We note in the introduction that in our paper we focus on accuracy rather than fidelity. The reasoning is twofold: first, fidelity-based extraction is not realistic since they assume very small ReLU-based networks; second, they are extremely inefficient even in such restricted settings.
> While fidelity is indeed an interesting objective for exploration, it has been shown by previous works [1,2,3] to underperform and therefore is out of scope for our work.
>
> In our work we focus on investigating the effect of having prior knowledge over the in-distribution region, and note that we make no assumptions about the input distribution itself. In our linear example, we provide intuition of the effect of an adversary that has no knowledge of what regions are of relevance and importance, and therefore require high accuracy, and what regions are of lesser importance and it is a waste of queries and effort to optimise the performance in these regions. The same argument can be sketched in a multi-dimensional setting. However, while in 2D the accuracy and fidelity arguments are inherently closely attached, this does not necessarily hold for higher dimensions.
>
>
> ```
> In Section 4.3, the authors discuss the ME performance gains from the use of surrogate datasets. However, they don't just use surrogate data alone; instead, they mix part of the original dataset with the surrogate data. It would be beneficial to understand the rationale behind this mixed approach.
> ```
> We investigate how essential it is for an attacker to have prior knowledge over the true data distribution. For this goal, we model prior knowledge as having actual access to real training samples. We mix the real training samples with the surrogate data samples, to model how the attack performance is affected by the availability of such prior knowledge.
>
>
>
> We hope this clarifies the reviewer’s concerns and ask the reviewer to please let us know if there are any other concerns we can address.
>
> [1] Matthew Jagielski, Nicholas Carlini, David Berthelot, Alex Kurakin, and Nicolas Papernot. High accuracy and high fidelity extraction of neural networks. In 29th USENIX security symposium (USENIX Security 20), pp. 1345–1362, 2020.
>
> [2] Nicholas Carlini, Matthew Jagielski, and Ilya Mironov. Cryptanalytic extraction of neural network models, 2020.
>
> [3] Adi Shamir, Isaac Canales-Martinez, Anna Hambitzer, Jorge Chavez-Saab, Francisco Rodrigez-Henriquez, and Nitin
> Satpute. Polynomial time cryptanalytic extraction of neural network models. arXiv preprint arXiv:2310.08708, 2023.

---

### Official Review · Reviewer_5YY3 · 2023-10-31

**Soundness:** 3 good
**Presentation:** 3 good
**Contribution:** 3 good
**Rating:** 5
**Confidence:** 4

**Summary:**

This paper conducts the measurement on how data access assumptions influence the accuracy of the extracted models, and how this changes adversary's incentives. Evaluation shows that victim labels leak limited information about the decision boundaries. The paper also discusses the possible defense based on the findings.

**Strengths:**

- Trendy topic
- New perspective for ME attacks

**Weaknesses:**

- More evaluation is needed
- Presentation can be improved
- Lack of certain theoretical support

**Questions:**

This paper presents the measurement results of model extraction attacks from a novel perspective. The authors argues that
with different access to the original training data, the performance of the model extraction attacks can be influenced. The paper
inspired researchers to think about model extraction attacks from another perspectives. However, it still has some flaws. I list my concerns in the followings:

- Based on my understanding, it is a measurement paper. Therefore, to achieve the conclusion, more experimental settings should be considered. For instance, if the architecture of the target models will influence the attack performance, or whether the target model that is trained under unsupervised loss may exhibit different results. I recommend the authors consider more settings for the comprehensive measurement conclusion.

- The writing needs to be improved. It seems that the warm up part is not necessarily connected to the following evaluation part. Also, in this paper, model stealing attacks and model extraction attacks are used confusingly. In the experimental results part, the main conclusion is also not that obvious.

- Besides the experimental results, a certain degree of theoretical proof is necessary. Note that the proof is not the illustration presented in the warm up part. I would suggest the authors explain the commonality of ME attacks more rigorously.

---

> ### Author Response · Authors · 2023-11-20
>
> We thank the reviewer for the detailed and insightful comments.
>
> ```
> Based on my understanding, it is a measurement paper. Therefore, to achieve the conclusion, more experimental settings should be considered. For instance, if the architecture of the target models will influence the attack performance, or whether the target model that is trained under unsupervised loss may exhibit different results. I recommend the authors consider more settings for the comprehensive measurement conclusion.
> ```
>
> In our paper we investigate the effect of data access assumptions over the success of query-based model extraction attacks. Although there are numerous attack papers, many of them share common methodology. Therefore, we carefully chose our test cases to best capture the similarities. This includes investigating both vision and NLP tasks, as well as investigating different types of surrogate datasets and query selection methods. As for the effect of different architectures - we performed preliminary analysis of the effect of the size of the model in Appendix I, and observed that it has no significant impact. We show that the victim model mostly serves as a labelling oracle, it is its accuracy (and correspondingly capacity) that matters for the attack’s performance, and not the exact architecture. This is also established in the knowledge distillation literature that shows how a model of different architecture and size can distil the knowledge of another model [1, 2]. This conclusion also applies for model’s trained under different loss functions - the victim model is labelling the queries according to its learned decision boundaries and is therefore leaking this information when queried with relevant samples.
>
> ```
> The writing needs to be improved. It seems that the warm up part is not necessarily connected to the following evaluation part. Also, in this paper, model stealing attacks and model extraction attacks are used confusingly. In the experimental results part, the main conclusion is also not that obvious.
> ```
> We thank the reviewer for this comment and will improve the clarity of the manuscript. We edited the last paragraph of Section 3 to clarify its connection to the evaluations that follow. We additionally replaced the term “model stealing” with “model extraction” to avoid any confusion.
>
> ```
> Besides the experimental results, a certain degree of theoretical proof is necessary. Note that the proof is not the illustration presented in the warm up part. I would suggest the authors explain the commonality of ME attacks more rigorously.
> ```
> Could the reviewer please clarify what kind of theoretical proof is missing? In addition to linear intuition, we discuss in Section 5.1-5.2 the hardness of sampling in-distribution in the absence of prior knowledge of it and connect it to the fundamental classification hardness.  We discuss how in the absence of the ability to distinguish OOD queries, the attacker inevitably faces a higher query complexity by being forced to sample from both regions. On the other hand, following the results of [4] we show that if the attacker can distinguish between the regions, it can already perform classification, given labels provided by the victim model or any other labelling oracle.
>
> As for our OOD component, we discuss the theoretical reasoning behind it in Section F.4. Moreover, a connection between ME and active learning has been established by [3], therefore posing a theoretical limit to what can be proved in this setting. In other words, anything general we can prove here will inevitably improve classical generalisation bounds, hence we chose to convey intuition using a toy linear setting instead.
>
>
> We hope this clarifies the reviewer’s concerns and ask the reviewer to please let us know if there are any other concerns we can address.
>
> [1] Jang Hyun Cho and Bharath Hariharan. On the efficacy of knowledge distillation, 2019
>
> [2] Paul Micaelli and Amos J Storkey. Zero-shot knowledge transfer via adversarial belief matching. In Advances in Neural Information Processing Systems, pages 9551–9561, 2019.
>
> [3] Varun Chandrasekaran, Kamalika Chaudhuri, Irene Giacomelli, Somesh Jha, and Songbai Yan. Exploring connections between active learning and model extraction. In 29th USENIX Security Symposium (USENIX Security 20), pp. 1309–1326, 2020.
>
> [4] Florian Tramer. Detecting adversarial examples is (nearly) as hard as classifying them, 2021.

---

### Official Review · Reviewer_7cgJ · 2023-10-31

**Soundness:** 2 fair
**Presentation:** 1 poor
**Contribution:** 2 fair
**Rating:** 3
**Confidence:** 3

**Summary:**

This paper studies model extraction attacks that aim to steal trained models with query access. It challenges the assumption that these attacks save on data costs, showing that they often rely on prior knowledge of in-distribution data, limiting their practical utility. In addition, the authors emphasize the need to separate prior knowledge from the attack policy and suggests a benchmark for evaluating attack policies directly.

**Strengths:**

- Studying model extraction attacks is a crucial avenue in the study of adversarial attacks, given the resource-intensive nature of training ML/DL models.
- The paper is easy to follow.

**Weaknesses:**

- While the authors draw conclusions from a diverse set of experiments, there appears to be a need for a principled approach to assess model extraction attacks. It would be valuable if the authors could provide clear and concise definitions that offer a unified perspective on all the attacks and defenses discussed in their paper. Currently, the experimental findings, while intuitive, seem somewhat fragmented and lack a well-organized presentation.
- Similar challenges are evident in the comparison between OOD and IND. If the authors could provide a clear and concise mathematical definition aligned with the objectives of model extraction attacks, it could offer deeper insights.

In summary, while the paper presents intriguing empirical findings, there is a need for the authors to establish straightforward and principled concepts to enhance the clarity of their arguments and conclusions.

**Questions:**

-

---

> ### Author Response · Authors · 2023-11-20
>
> We thank the reviewer for the comments.
> ```
> While the authors draw conclusions from a diverse set of experiments, there appears to be a need for a principled approach to assess model extraction attacks. It would be valuable if the authors could provide clear and concise definitions that offer a unified perspective on all the attacks and defenses discussed in their paper. Currently, the experimental findings, while intuitive, seem somewhat fragmented and lack a well-organized presentation.
> ```
> Model extraction attacks in the current literature practically perform knowledge distillation. That in turn means that classical learning generalisation bounds apply and no general theory can be developed.
>
> In the same direction, [1, USENIX] argued that ``the process of ME is very similar to active learning’’ and therefore suggested that any improvement in active learning translates to improvement in ME attacks. In our work we address the common assumption that ME poses a significant threat (see [2]) and identify the reason why existing attacks “work”. We discover this success is mostly the result of having some prior knowledge, and with the absence of such prior the attack reduces to providing a labelling oracle for the attacker’s unlabeled data. As such we call for the community to re-evaluate the threat and perform more appropriate threat modelling.
>
> We are certain that stronger attacks will appear and it is paramount that we are capable of telling when they are discovered and why such attacks work. Our paper does the first step in that direction and separates away the impact of data prior and the attack policy.
>
> ```
> Similar challenges are evident in the comparison between OOD and IND. If the authors could provide a clear and concise mathematical definition aligned with the objectives of model extraction attacks, it could offer deeper insights.
> ```
> We do model the OOD adversaries and connect with the existing literature in our Section 5.2, which allows us in many cases to formally state that extracting the model is as hard as solving the classification task at hand. At the same time, we do want to also note that the IND/OOD community also has no clear definitions of what the distinctions are here.
>
> We hope this clarifies the reviewer’s concerns and ask the reviewer to please let us know if there are any other concerns we can address.
>
> [1] Varun Chandrasekaran, Kamalika Chaudhuri, Irene Giacomelli, Somesh Jha, and Songbai Yan. Exploring connections between active learning and model extraction. In 29th USENIX Security Symposium (USENIX Security 20), pp. 1309–1326, 2020.
>
> [2] Ram Shankar Siva Kumar, Magnus Nyström, John Lambert, Andrew Marshall, Mario Goertzel, Andi Comissoneru, Matt Swann, and Sharon Xia. Adversarial machine learning-industry perspectives. In 2020 IEEE Security and Privacy Workshops (SPW), pp. 69–75. IEEE, 2020.

---

> > ### Comment · Reviewer_7cgJ · 2023-11-22
> > **Thanks for the authors' rebuttal**
> >
> > I appreciate the authors for addressing my concerns, but I don't feel that my worries have been eased.
> >
> > For example, you said
> > > we do want to also note that the IND/OOD community also has no clear definitions of what the distinctions are here.
> >
> > While the above observations may apply to the broader OOD community, they should not discourage you from providing precise and unified definitions. As seen in [A], the authors meticulously define OOD and outline various properties. While their definitions may not be universally accepted, they do offer both quantitative and qualitative analyses, at least from my perspective.
> >
> > Thus, I will maintain my current rating.
> >
> > [A]: Fang et al., "Is Out-of-Distribution Detection Learnable?", in NeurIPS 2022.

---

### Official Review · Reviewer_HWBR · 2023-11-01

**Soundness:** 2 fair
**Presentation:** 3 good
**Contribution:** 2 fair
**Rating:** 5
**Confidence:** 4

**Summary:**

The paper investigates query-based model extraction attacks, where the objective is to train a substitute model that exhibits similar performance to the stolen one. The authors argue about the real advantages of performing this kind of attack, especially considering the requirement of a surrogate dataset from the same distribution of the victim training data. Moreover, they propose an OOD-based approach, that avoids unveiling the actual model decision boundaries to out-of-distribution queries, assuming the attacker mostly relies on them to build its surrogate dataset, as a benchmark to evaluate how the availability of data influences the attack success.

**Strengths:**

The literature on this topic is scattered, and it is very difficult to have a clear assessment of the real-world impact of model extraction attacks, as the considered settings can be very different in their assumptions (knowledge of model architecture, training data distribution, preprocessing steps, output access, application domain, etc.). Often, even comparing different attacks and defenses is not straightforward. This work has the merit of trying to address some of these issues.

**Weaknesses:**

Although the considerations and the findings of the paper are very interesting, its contribution seems limited by the considered case studies, whereas the conclusions drawn by the authors are supposed to be generally applied.

Model stealing attacks can be performed with different purposes and settings, but in this work, the analyzed ones and experimental evaluation include a very tiny set of them: for instance, the attacker might easily obtain some data from the training distribution (or similar ones), a pre-trained model might be already available, in some domains it is not possible to use automatic labeling services, and in general the cost of query the victim model and make a sample labeled by a third-party service might widely vary.
Several attacks cover these or other settings (see [a, b]) and some of them seem very practical in a real-world scenario. In addition, as this research field is relatively recent, some of the authors' claims might become less strong as new more efficient attacks are designed: for instance, a recent work [c] shows that there is still room for improvements in this sense.

Regarding the evaluation using additional data samples from different distributions: I don't expect that simply using them to train the substitute model after being labeled from the victim can give high improvements, whereas several attacks use them as a starting point to build a more efficient surrogate training set with some optimization technique (similar to active learning). The authors should have considered these attacks to evaluate how the availability of these data influences the attack's success. A similar consideration can be made with respect to the knowledge of a very small fraction of the training data.

[a] Oliynyk, D., Mayer, R., & Rauber, A. (2022). I Know What You Trained Last Summer: A Survey on Stealing Machine Learning Models and Defences. ACM Computing Surveys, 55, 1 - 41.
[b] Yu, H., Yang, K., Zhang, T., Tsai, Y., Ho, T., & Jin, Y. (2020). CloudLeak: Large-Scale Deep Learning Models Stealing Through Adversarial Examples. Network and Distributed System Security Symposium.
[c] Lin, Z., Xu, K., Fang, C., Zheng, H., Ahmed Jaheezuddin, A., & Shi, J. (2023). QUDA: Query-Limited Data-Free Model Extraction. Proceedings of the 2023 ACM Asia Conference on Computer and Communications Security.

**Questions:**

Can you please extend the overall discussion and the experiments to a wider range of settings and attacks?

---

> ### Author Response · Authors · 2023-11-20
>
> We thank the reviewers for the detailed review.
>
> ```
> Although the considerations and the findings of the paper are very interesting, its contribution seems limited by the considered case studies, whereas the conclusions drawn by the authors are supposed to be generally applied.
> ```
>
> It is indeed the case that our paper is empiric and analyses performance of different attacks. Having said that, we disagree with the premise that conclusions are overstating the findings. Our conclusions state that the performance of current attacks in the literature are primarily driven by prior knowledge for the settings of 1%+ of the original training dataset; we then take this commonly used setting to the limit – our OOD evaluation clearly demonstrates how, in presence of no prior knowledge, both theoretically and practically no attacks outperform random querying over the whole input domain.
>
> At the same time, Chandrasekaran et al (2020, USENIX) demonstrated that model extraction is equivalent to active learning, suggesting that if general theory were to be derived here, it would essentially be the same as for general machine learning and generalisation. We refer the reviewer to Appendix B where we cover it all in detail.
>
> We chose our evaluated case studies such that they represent the common methodologies in ME literature, and thus enable us to derive a general conclusion. We discuss this in more detail below, when addressing the reviewer’s question about the use of surrogate datasets.
>
> ```
> Model stealing attacks can be performed with different purposes and settings, but in this work, the analyzed ones and experimental evaluation include a very tiny set of them: for instance, the attacker might easily obtain some data from the training distribution (or similar ones), a pre-trained model might be already available, in some domains it is not possible to use automatic labeling services, and in general the cost of query the victim model and make a sample labeled by a third-party service might widely vary. Several attacks cover these or other settings (see [a, b]) and some of them seem very practical in a real-world scenario. In addition, as this research field is relatively recent, some of the authors' claims might become less strong as new more efficient attacks are designed: for instance, a recent work [c] shows that there is still room for improvements in this sense.
> ```
> In our paper we cover the most commonly used model stealing setting. Although attacks do get better with time and more prior is clearly useful, both of the points do not argue against our paper. Our paper states that *current attacks* are benefiting from prior knowledge rather than the query policy introduced by the attack. Stronger attacks with better use of prior should certainly exist and our paper suggests that they should be evaluated with great care, where impact of prior is detached from the attack policy itself. That is precisely why we wrote the paper in the first place.
>
> [c] mentioned above is a clear example of why our paper is important. Does [c] improve over DFME in its handling of prior information or is the improvement because the attack policy is better?

---

> ### Author Response · Authors · 2023-11-20
>
> ```
> Regarding the evaluation using additional data samples from different distributions: I don't expect that simply using them to train the substitute model after being labeled from the victim can give high improvements, whereas several attacks use them as a starting point to build a more efficient surrogate training set with some optimization technique (similar to active learning). The authors should have considered these attacks to evaluate how the availability of these data influences the attack's success. A similar consideration can be made with respect to the knowledge of a very small fraction of the training data.
> ```
> What attacks does the reviewer have in mind that follow this process? Does the reviewer have a reason to believe that the performance of the attack will differ from the ones covered in the current manuscript?
>
> In the paper we follow a standard process of launching model stealing attacks. The common methodology in query-based model extraction attacks, as is also described in the survey the reviewer referred to [a] (see Section 7.5), is for an attacker to (1) obtain some unlabelled dataset; (2) label it by querying the victim model; and (3) use this data to train a substitute model. Usually attacks also assume additional control, for example they assume control over
> - the distribution of the surrogate dataset,
> - the sampling of the dataset,
> - the availability of prior knowledge over the true distribution
> - pretraining, etc.
>
> As noted in our paper, as well as in [a], different works differ in the type of queries they use and in the way they choose which queries to issue. Our experiments, run over months, thoroughly and comprehensively cover a range of different settings to allow apples-to-apples comparison between attacks. This include different levels of prior knowledge, different types of surrogate datasets (random and nonsensical, similar distribution, different distribution, and carefully generated), different query ordering (random, reinforcement learning (KnockoffNets), and exploration-based (DFME)), pretraining (KnockoffNets’ models and the MNLI models are pre-trained), different access levels (soft labels, hard labels), etc. In all of our experiments, we have reached the same conclusions, aligning with the theoretical intuition - the biggest effect over the attack success is related to the level of prior knowledge (or the use of a large query budget as in DFME).
>
> Thank you very much for pointing [c] out, the implementation is not publicly available and it is not feasible to obtain all results for it in time for the rebuttal. As [c] suggests improvement over DFME (i.e. pretrained GAN and adversarial perturbations) it is reasonable to believe that it will achieve better performance than DFME did in our evaluations. However, although the exact results would differ, [c] still traverses the input space and will be affected by limiting the informativeness of OOD queries (in particular, the adversarial perturbation might ‘’push’’ some queries to this region although starting from IND samples.
>
>
> The reviewer mention that other works make more sophisticated use of the surrogate dataset - we are not aware of any other way to improve the use of the dataset then to order it in a different way, which according to our results as well as the results presented in the KnockoffNets paper, does not have a big impact. Can the reviewer direct us to a paper that makes a different use of the available data?
>
> Regarding the use of a very small fraction of the data - we have evaluated the effect of having different amounts of data, including having as little as 1% of the data and we make practical evaluation suggestions for future work.
>
> We hope this clarifies the reviewer's concerns and ask the reviewer to raise any additional concerns we can address.
>
>
> [1] Varun Chandrasekaran, Kamalika Chaudhuri, Irene Giacomelli, Somesh Jha, and Songbai Yan. Exploring connections between active learning and model extraction. In 29th USENIX Security Symposium (USENIX Security 20), pp. 1309–1326, 2020.

---

> > ### Comment · Reviewer_HWBR · 2023-12-04
> > **Response to the rebuttal**
> >
> > Dear authors,
> >
> > thanks for your thorough clarifications, but although the motivations that drive your work are more clear to me, my main concerns remain.
> > In particular, I think that more attacks that leverage techniques to augment and optimize the surrogate training data should be considered, such as, for instance, Black-Box Ripper, ActiveThief, ES Attack and InverseNet.

---

### Meta-Review · Area_Chair_HN1L · 2023-12-03

**Metareview:**

This paper studies model-extraction (ME) attacks that aim to recover a victim model using black-box queries. It questions the practical value of ME attacks from two perspectives.

The first perspective is labeling cost. If the ME adversaries can access the original training data, the victim model cannot perform much better than a labeling oracle (even if soft labels are used). Since labeling cost is low thanks to crowdsourcing, ME attacks are not useful. If the ME method can access extra data from a surrogate dataset, unless the extra data is very close to the original (e.g., Cifar 100 to Cifar10), using the surrogate data does not help much when the ME adversaries have some prior knowledge of the original data.

The second perspective is OOD, especially how much the ME attacks rely on the assumption that OOD queries provide ID information. According to the authors, the implicit assumption of previous work is that IND decision boundary can be inferred from the OOD ones. If this assumption does not hold:
- If adversaries can tell IND from OOD, they will just query IND. This boils down to the labeling oracle argument.
- If adversaries cannot tell IND from OOD, they will need to solve the OOD detection first, which itself is a nontrivial problem and is not easier than performing classification itself.

The authors then designed a way to make OOD data useless in inferring the IND decision boundaries. They showed that this method makes the ME attack much less effective.

**Justification For Why Not Higher Score:**

After reading the paper, I am slightly surprised that it received such low scores. I will mainly summarize the reviewers' comments and provide my understanding.

Reviewers pointed out several problems of the paper:
- The contributions seem limited by the considered case studies. Different factors and varying environments should be considered.
- Limited discussions on newer and stronger attacks.
- Lack of a unifying framework for defining attacks and defenses. Lack of mathematical definitions. The different experiments in the paper seem fragmented.
- The linear classifier example is not aligned with later discussions on classification models.
- The rationale for mixing original training data with surrogate data (in Section 4.3) needs explanation.
- Paper writing and presentation issues.

Some of my understanding:
- The clarity can indeed be improved. Currently, the information in the paper is quite dense, and some discussions are abrupt. Also, due to the lack of math definitions, it often takes me several times to read a paragraph to guess what it means.
- The paper's result is somewhat "expected." It is not the type of surprising results that ML audience likes. For example, if I can request the victim's model using a subset D of training data, and the victim model does not differ significantly in size from the recovered model, why should I expect the performance of ME attacks to be better than training with D? In other words, why should I expect ME attacks to perform better than a labeling oracle?
- This paper does not provide a clear picture of the labeling cost. While it may be true that labeling is cheap in some cases, the positioning of the paper seems to suggest it is cheap to the point that providing labels is useless. To me, being able to get a labeling oracle is not a bad thing.
 - To some extent, the paper assumes too much from the reviewers. The theoretical reviewers would probably require the paper to use rigorous math language. The empirical reviewers would probably like more discussions and connections to newer, stronger ME attack schemes.

However, I believe this is a good paper. The authors should consider revising it and submitting the paper to future venues.

**Justification For Why Not Lower Score:**

N/A

---

### Decision · Program_Chairs · 2024-01-16

Reject